JCB | Journal of Cell Biology

# Lipid exchange at ER–trans-Golgi contact sites governs polarized cargo sorting

Dávid Kovács[1], Anne-Sophie Gay[1], Delphine Debayle[1], Sophie Abélanet[1], Amanda Patel[1], Bruno Mesmin[1], Frédéric Luton[1], and Bruno Antonny[1]

Oxysterol binding protein (OSBP) extracts cholesterol from the ER to deliver it to the TGN via counter exchange and subsequent hydrolysis of the phosphoinositide PI(4)P. Here, we show that this pathway is essential in polarized epithelial cells where it contributes not only to the proper subcellular distribution of cholesterol but also to the trans-Golgi sorting and trafficking of numerous plasma membrane cargo proteins with apical or basolateral localization. Reducing the expression of OSBP, blocking its activity, or inhibiting a PI4Kinase that fuels OSBP with PI(4)P abolishes the epithelial phenotype. Waves of cargo enrichment in the TGN in phase with OSBP and PI(4)P dynamics suggest that OSBP promotes the formation of lipid gradients along the TGN, which helps cargo sorting. During their transient passage through the trans-Golgi, polarized plasma membrane proteins get close to OSBP but fail to be sorted when OSBP is silenced. Thus, OSBP lipid exchange activity is decisive for polarized cargo sorting and distribution in epithelial cells.

## Introduction

Epithelial cells line the surfaces of our body and are differentiated to form tightly connected cellular sheets that act as biological barriers. As a consequence of tight junction formation, the epithelial plasma membrane is divided into distinct apical and basolateral domains, which are equipped with different sets of proteins and lipids. While glycosylphosphatidylinositol (GPI)-anchored proteins are mostly present on the apical surface, cadherins and integrins are located on the basolateral plasma membrane domain to form adherent junctions and focal adhesions, respectively (Paladino et al., 2006; Caceres et al., 2019; Keller et al., 2001; Lebreton et al., 2019). Additionally, compared with the basolateral plasma membrane, the apical membrane is more ordered and enriched in cholesterol, demonstrating the polarized distribution of membrane lipids within the epithelial plasma membrane (Gerl et al., 2012).

To achieve and maintain this cell polarity, cargoes are sorted in the trans-Golgi network (TGN) and directed toward the apical and basolateral membranes using distinct secretory routes. Although TGN sorting signals and their recognizing machinery have been well-characterized, the molecular mechanisms of clustered cargo trafficking are still largely unknown (Ramazanov et al., 2021; Boncompain and Weigel, 2018). It has recently been suggested that cargo-specific sorting and subsequent trafficking routes are primarily regulated by the dynamic adaptation of the TGN to the particular cargo (Boncompain and Weigel, 2018). Furthermore, apical and basolateral cargoes are

able to segregate from each other and follow distinct cognate routes even in non-epithelial cells, suggesting that apicobasal cargo sorting is not limited to polarized cells and might be governed by the cargo itself (Yoshimori et al., 1996).

It has long been recognized that the lipid environment affects protein sorting and trafficking (Keller and Simons, 1997, 1998). Lipid composition defines the biophysical properties of the membranes, which might facilitate the formation of different TGN subdomains and the recruitment of specific groups of proteins leading to the assembly of cargo-specific machinery (von Blume and Hausser, 2019; Klemm et al., 2009). For instance, apically sorted GPI-anchored proteins are clustered in cholesterol-rich membrane regions in the TGN, which drives their lateral segregation from other cargo types (Paladino et al., 2004; Simons and Ikonen, 1997; Zurzolo and Simons, 2016). However, our understanding of the link between lipid environment and cargo sorting has remained fragmented due to the difficulty in tuning the lipid composition of organelles both specifically and locally.

Recent advances in the characterization of membrane contact sites (MCS) offer new strategies to study the impact of lipid composition on membrane traffic steps. MCS are regions of close apposition between organelles, which are enriched in lipid transfer proteins (Wu et al., 2018). Oxysterol-binding protein (OSBP)—a member of the OSBP-related protein family (ORPs)—is recruited to the ER-TGN MCS to transfer cholesterol

[1]Université Côte d'Azur and CNRS, Institut de Pharmacologie Moléculaire et Cellulaire, Valbonne, France.

Correspondence to Bruno Antonny: antonny@ipmc.cnrs.fr.

from the endoplasmic reticulum (ER) to the TGN, thereby controlling the lipid composition of trans-Golgi membranes (Antonny et al., 2018; Mesmin et al., 2019). This transfer activity requires the metabolic energy of the lipid phosphatidylinositol-4-phosphate (PI(4)P), which has an unequal distribution between the ER and TGN (Antonny et al., 2018; Mesmin et al., 2013, 2017). Due to the activity of TGN-resident PI4-kinases, PI4KIIα and PI4KIIIβ, the PI(4)P concentration is high in the TGN, while it is subjected to degradation in the ER by the phosphatase Sac-1. When OSBP transfers a cholesterol molecule to the TGN, it extracts one PI(4)P from the TGN and transfers it back to the ER. As such, ER to TGN forward transfer of cholesterol occurs owing to the backward transfer and subsequent hydrolysis of PI(4)P (Mesmin et al., 2013, 2017).

Because cholesterol is recognized as a major lipid that controls cargo trafficking (von Blume and Haussser, 2019; Dukhovny et al., 2009; Sugiki et al., 2012), we surmised that OSBP should affect the sorting and the subsequent trafficking of cargoes that depend on the local amount of this lipid. Here, using various proteomics and cell biology approaches, we report that OSBP regulates polarized cargo trafficking in epithelial cells. We found that the post-Golgi trafficking of numerous apical and basolateral proteins depends on OSBP. Moreover, live-cell experiments suggest that OSBP dynamics at the TGN membranes drive the sorting of cargoes with polarized distribution. These observations provide a better understanding of cargo sorting for epithelial polarity, which has multiple impacts on pathological aspects as well, notably in cancers of epithelial origins.

## Results

### OSBP dynamics drive apicobasal cargo sorting and polarity establishment

We previously reported that OSBP contact sites are highly dynamic in hTERT-RPE1 cells—a cell line with a well-developed TGN, ideal for real-time imaging. They show traveling waves that are sensitive to changes in PI(4)P synthesis as well as to perturbations of membrane saturation and protein crowding (Mesmin et al., 2017; Jamecna et al., 2019). Notably, gradual inhibition of PI4KIIIβ by PIK93 increases the amplitude of these waves, reflecting a tug-of-war between PI(4)P synthesis by two PI4-kinases (PI4KIIα and PI4KIIIβ) and OSBP-dependent consumption of PI(4)P (Mesmin et al., 2017). We aimed to determine whether cell surface cargoes in the TGN follow the OSBP oscillations by exploiting the Retention Using Selective Hooks (RUSH)-system (Boncompain et al., 2012). For this, we coexpressed a RUSH plasmid expressing EGFP-GPI together with the PI(4)P probe mCherry-PH$_{OSBP}$, which can be used as a reporter of OSBP contact sites in hTERT-RPE1 cells (Mesmin et al., 2013). We released EGFP-GPI from the ER by biotin addition and then enhanced the PI(4)P waves by adding PIK93 when the cargo started to accumulate at the TGN. Strikingly, we found that Golgi release of EGFP-GPI followed the dynamics of OSBP oscillations as we observed that the curves corresponding to mCherry-PH$_{OSBP}$ and EGFP-GPI were in phase (Fig. 1, A–C). This observation suggests that TGN dynamics of surface cargoes are synchronized with the OSBP cycle.

Next, we aimed to determine whether OSBP dynamics affect apicobasal cargo sorting as well. For this, we used the epithelial cell line MDCK, which showed noticeable PI(4)P waves with high amplitudes already at steady state (Fig. 1, D and E). In these cells, the amplitude of the PI(4)P waves reported by the mCherry-PH$_{OSBP}$ probe diminished upon PIK93 addition, probably due to the dissociation of OSBP from the TGN membranes (Fig. 1, D and E; and Fig. S1 A). We exploited this effect of PI4KIIIβ inhibition in MDCK cells to test the contribution of PI(4)P dynamics to apicobasal cargo sorting. We cotransfected MDCK cells with RUSH constructs expressing apically sorted EGFP-GPI and basolateral mCherry-CDH1 model cargoes. Following the induction of cargo release by biotin addition, we synchronized the two cargoes at the TGN by a 1.5 h temperature block at 19.5°C. Following this step, cells were either fixed directly or further incubated at 37°C for 40 min with or without PIK93 before fixation to let the cargoes leave the TGN. After the temperature block, EGFP-GPI and mCherry-CDH1 colocalized at the TGN (Fig. 1 F). Following a 40 min incubation at 37°C, EGFP-GPI and mCherry-CDH1 in the DMSO-treated control cells were detected mostly in distinct post-Golgi vesicles, indicating that the two cargo types had segregated from each other (Fig. 1, F and G). When the cells were treated with PIK93 after the temperature block, the percentage of post-Golgi vesicles containing both cargoes increased significantly, indicating a defect in cargo sorting (Fig. 1, F and G).

We also assessed the effect of PI4KIIIβ inhibition on the development of MDCK cysts. When MDCK cells were left to develop cysts in the Matrigel matrix for 72 h, most of the cells from the DMSO-treated control sample developed cysts containing a single central lumen, indicating an efficient polarity establishment. In the presence of PIK93, however, a significantly higher number of cysts with luminal defects were detected. Typically, these cysts displayed multiple lumens and contained mislocalized surface cargoes, suggesting a defect in polarized cargo sorting (Fig. 1, H and I; and Fig. S1 B).

Because PIK93 is reported to inhibit some PI3Ks as well, we repeated these experiments using a more specific inhibitor PI4KIIIbeta-10-IN (Knight et al., 2006). Similar to PIK93, this compound triggered the partial dissociation of OSBP from the TGN membrane (Fig. S1 C) and impaired apicobasal RUSH cargo sorting (Fig. S1, D and E). Furthermore, the presence of PI4KIIIbeta-10-IN significantly increased the frequency of cysts with morphological defects (Fig. S1, F and G), confirming the importance of PI(4)P metabolism in cargo sorting.

Next, we investigated whether OSBP silencing also leads to apicobasal sorting defects. After confirming efficient silencing (Fig. S1 H), we tested the dynamics of CDH1 and EGFP-GPI trafficking upon OSBP depletion. Both cargoes showed similar dynamics upon siOSBP treatments (Fig. 1, J and K; and Fig. S1 I). In control cells nucleofected with siNT, the cargoes typically reached the TGN within 30–40 min after ER release; thereafter, they left the TGN and appeared at the cell surface. Although cargoes in OSBP-silenced cells were able to leave the ER, we observed significant delays in the post-Golgi trafficking, indicating that OSBP activity is necessary for efficient cargo release from the TGN (Fig. 1, J and K; and Fig. S1 I).

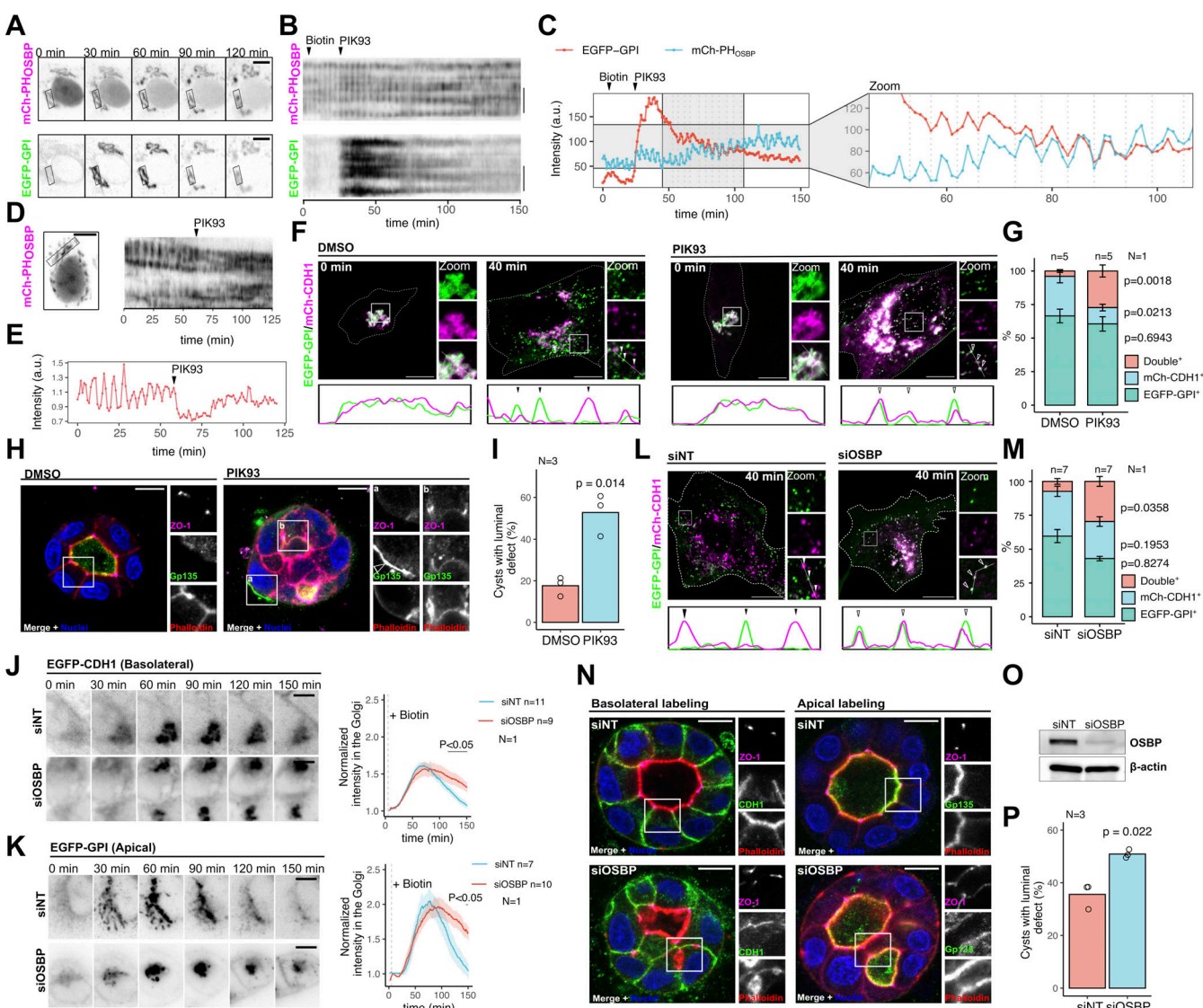

Figure 1. **OSBP dynamics regulate apicobasal cargo sorting. (A–C)** hTERT-RPE1 cells were cotransfected with EGFP-GPI-expressing RUSH plasmid and mCherry-PH$_{OSBP}$ construct. Epifluorescence time-lapse imaging was performed the next day. Biotin was added to the cells to initiate cargo release at 5 min, and PIK93 was applied 30 min after the start of the experiment at 500 nM. Framed Golgi areas indicate regions that were used to generate kymographs shown in B. Intensity values corresponding to the indicated spots of the Golgi area over time were plotted on C. Bar = 10 μm. **(D and E)** MDCK cells were nucleofected to express mCherry-PH$_{OSBP}$. Epifluorescence time-lapse imaging was performed on the next day. Kymograph corresponding to the indicated Golgi area (D) and Golgi fluorescence intensity values over time were plotted (E). PIK93 was added to the cells at the indicated time point at 500 nM. Bar = 10 μm. **(F and G)** MDCK cells were cotransfected with EGFP-GPI and mCherry-CDH1 expressing RUSH constructs, and cargo release was triggered by biotin addition on the next day. Following biotin addition, cells were kept at 19.5°C for 1.5 h to synchronize cargoes at the TGN. Cells were released from the temperature block, fixed, or kept in the presence of DMSO or 500 nM PIK93 at 37°C for 40 min followed by fixation and confocal microscopy. Line-plots indicate the fluorescence intensities of each cargo at the marked areas. Arrowheads point to post-Golgi vesicles positive for single or double cargoes (F). Distribution of each single- and double cargo-containing post-Golgi vesicle in control and PIK93-treated cells. N = number of independent experiments presented, n = number of cells/sample used for the analysis. In each cell, at least 50 vesicles were counted. Error bars show SEM, and P values were calculated by two-way ANOVA followed by Sidak's multiple comparison test. Bar = 10 μm, diameter of the insets = 8.7 μm. **(H and I)** MDCK cells were seeded in Matrigel and left to form cysts for 72 h in the presence of DMSO or 500 nM PIK93. Cysts were then fixed, stained, and analyzed by confocal microscopy. The arrowhead indicates Gp135 facing the extracellular matrix (H). PIK93 treatment increases the percentage of MDCK cysts with morphology defects. More cysts are shown in Fig. S1 B. N = number of independent experiments presented. Statistics were obtained by counting at least 70 cysts/condition. P value was calculated by unpaired t test. Bar = 10 μm, diameter of the insets = 14.5 μm. **(J and K)** MDCK cells were nucleofected with control or OSBP-silencing siRNA. 24 h later, cells were transfected again with basolateral CDH1 (J) or apical EGFP-GPI (K) RUSH model cargoes. Epifluorescence time-lapse microscopy was performed on the next day and fluorescence intensity values corresponding to Golgi areas were plotted. Bold lines with the shaded areas indicate mean ± SEM. P values were calculated by two-way ANOVA followed by Sidak's multiple comparison test. N = number of independent experiments presented. n = number of cells used to calculate mean and error. The three independent experiments are shown in Fig. S1 I. Bars = 10 μm. **(L and M)** MDCK cells were nucleofected with non-targeting or OSBP-silencing siRNA. 24 h later, cells were cotransfected with basolateral CDH1 and apical EGFP-GPI model cargo-expressing constructs. On the next day, biotin was added, and then cells were kept at 19.5°C for 1.5 h to synchronize cargoes at the TGN. Following this, cells were kept at 37°C for 40 min, followed by fixation and visualization by confocal microscopy. Line-plots indicate fluorescence intensities of each cargo at the corresponding areas and arrowheads point to post-Golgi

vesicles positive for single or double cargoes (L). Distribution of each single- and double cargo-containing post-Golgi vesicle in control and PIK93-treated cells (M). Seven cells/condition were quantified; in each cell, at least 30 post-Golgi vesicles were counted. N = number of independent experiments presented, n = number of cells/sample used for the analysis. Error bars show SEM and P values were calculated by two-way ANOVA followed by Sidak's multiple comparison test. Bar = 10 µm, diameter of the insets = 8.7 µm. **(N–P)** MDCK cells were nucleofected with control siRNA or siOSBP, seeded onto Matrigel 24 h later, and left to develop cysts for 72 h. Representative cysts stained with apical/basolateral markers are shown in N. More cysts are shown on Fig. S1 J. Efficacy of RNAi-based OSBP silencing was tested 4 d post-nucleofection in MDCK cells (O). OSBP silencing increases the number of MDCK cysts with morphology defects. Each experiment was done in triplicate/condition and statistics were obtained by counting at least 70 cysts/condition. P value was calculated by unpaired t test. N = number of independent experiments presented. Bar = 10 µm, diameter of the insets = 14.5 µm. Source data are available for this figure: SourceData F1.

To test whether this delay in transport is coupled to sorting defects as well, we quantified EGFP-GPI and mCherry-CDH1-containing post-Golgi vesicles following temperature block release in control siRNA-transfected and OSBP-silenced cells. The number of post-Golgi vesicles containing both cargo types significantly increased upon OSBP silencing, indicating that OSBP activity contributes to the segregation of apical and basolateral cargoes (Fig. 1, L and M).

Finally, we compared the morphology of 3D-polarized cysts developed from control and OSBP-silenced MDCK cells. While only ~35% of the cysts showed irregular morphology in the control siRNA-nucleofected cells, ~50% of the OSBP-silenced cyst population showed luminal morphology defects. Typically, these cysts developed multiple lumens and displayed apical multipolarity where one cell formed several apical surfaces, presumably due to the mis-sorting of apical cargoes (Fig. 1, N–P; and Fig. S1 J).

We concluded that the OSBP/PI(4)P cycle at TGN membranes drives the segregation of cargoes and, as such, governs their proper trafficking to establish epithelial cell polarity.

## OSBP inhibition blocks post-Golgi trafficking and disrupts epithelial polarity

Because OSBP regulates apicobasal sorting and cargo exit at the TGN, we assessed whether polarized cargoes approach OSBP-containing MCSs during their Golgi trafficking. For this aim, we coexpressed four model RUSH cargoes with various topologies and final destinations with the catalytically inactive form of OSBP, mCherry-OSBP$^{HHK>AAA}$, which can be used as a passive reporter for OSBP at the contact sites. These chosen cargoes were CDH1–E-cadherin (basolateral), DSG2–Desmoglein2 (non-polarized), EGFP–GPI (apical), and GP135–Podocalyxin Like (apical). We initiated their release from the ER by biotin addition; then cells were subsequently kept at 19.5°C for 1.5 h to accumulate cargoes in the trans-Golgi. Following this, samples were switched back to 37°C to monitor cargo export dynamics by live imaging. We exploited the blocking effect of the ORPphilin OSW-1 on OSBP. Mechanistically, ORPphilins—such as OSW-1 and SWG—inhibit OSBP-mediated lipid exchange, thereby increasing the level of PI(4)P at the TGN, which results in the forced recruitment and subsequent stabilization of inactive OSBP at contact sites (Mesmin et al., 2017).

At 19.5°C, cargoes colocalized with mCherry-OSBP$^{HHK>AAA}$, indicating their accumulation at the TGN. In control cells, the shift to 37°C released the cargoes from the TGN, while mCherry-OSBP$^{HHK>AAA}$ still localized to the TGN. Treatments with OSW-1 strongly impaired TGN export of the CDH1 and DSG2 cargoes, as demonstrated by a persistent co-localization with OSBP over

time (Fig. 2, A and B; and Fig. S2, A and B). In contrast, the apically-sorted proteins EGFP-GPI and GP135 were able to leave the Golgi, albeit at a slower rate than in controls (Fig. 2, C and D; and Fig. S2, C and D). As expected, OSW-1 treatments triggered mCherry-OSBP$^{HHK>AAA}$ recruitment to the Golgi due to inhibition of PI(4)P turnover (Fig. 2, A–D; and Fig. S2, A–D). These colocalization experiments suggest that both apical and basolateral cargoes require OSBP-dependent MCSs during their transit at the Golgi apparatus; however, basolateral model cargoes appear more sensitive to OSW-1.

Next, we tested the effect of OSBP inhibition on MDCK cysts. Strikingly, overnight treatment of polarized MDCK cysts with the OSBP-blocking drug SWG led to a drastic loss of cell polarity as the lumens disappeared and the cysts acquired a grape-like morphology consisting of rounded cells showing no asymmetrical marker distribution (Fig. 2 E and Fig. S2 E). Interestingly, we detected less E-cadherin and β-catenin signals at cell–cell junctions in SWG-treated cysts. Additionally, GP135 was detected in plasma membrane domains facing the extracellular matrix as well (Fig. S2 E).

We tested the effect of SWG on MDCK cells under conditions where their polarity is minimal; that is when they form 2D colonies. To observe morphology changes of MDCK colonies, we performed overnight time-lapse imaging after the addition of DMSO or the OSBP inhibitor SWG (Fig. 2 F and Video 1). In the DMSO-treated control, we observed basal expansion of the colonies due to cell proliferation; however, after a few hours of SWG treatment, the colonies started to scatter, indicating that OSBP inhibition perturbs cell–cell and cell–substrate anchoring functions.

Altogether, these results indicate that OSBP-engaged MCSs govern apical and basolateral sorting routes and contribute to the maintenance of epithelial polarity.

## Proximity ligation reveals apicobasal cargo segregation close to OSBP

One caveat of the RUSH assay is to rely on overexpressed cargoes, which might saturate the endogenous sorting machinery of the cell and could lead to sorting defects. Therefore, we aimed to determine the repertoire of endogenous cargoes that transiently become close to OSBP during their sorting. For this, we chose a proximity ligation strategy in which the promiscuous biotin ligase TurboID was fused to the C-terminus of OSBP (Fig. S3 A). Upon biotin addition, OSBP-TurboID biotinylates proteins close to OSBP. Following pull-down with streptavidin beads, we identified these proteins by mass spectrometry (Branon et al., 2018).

We performed the OSBP-TurboID assays either in the absence or in the presence of ORPphilins to take advantage of the

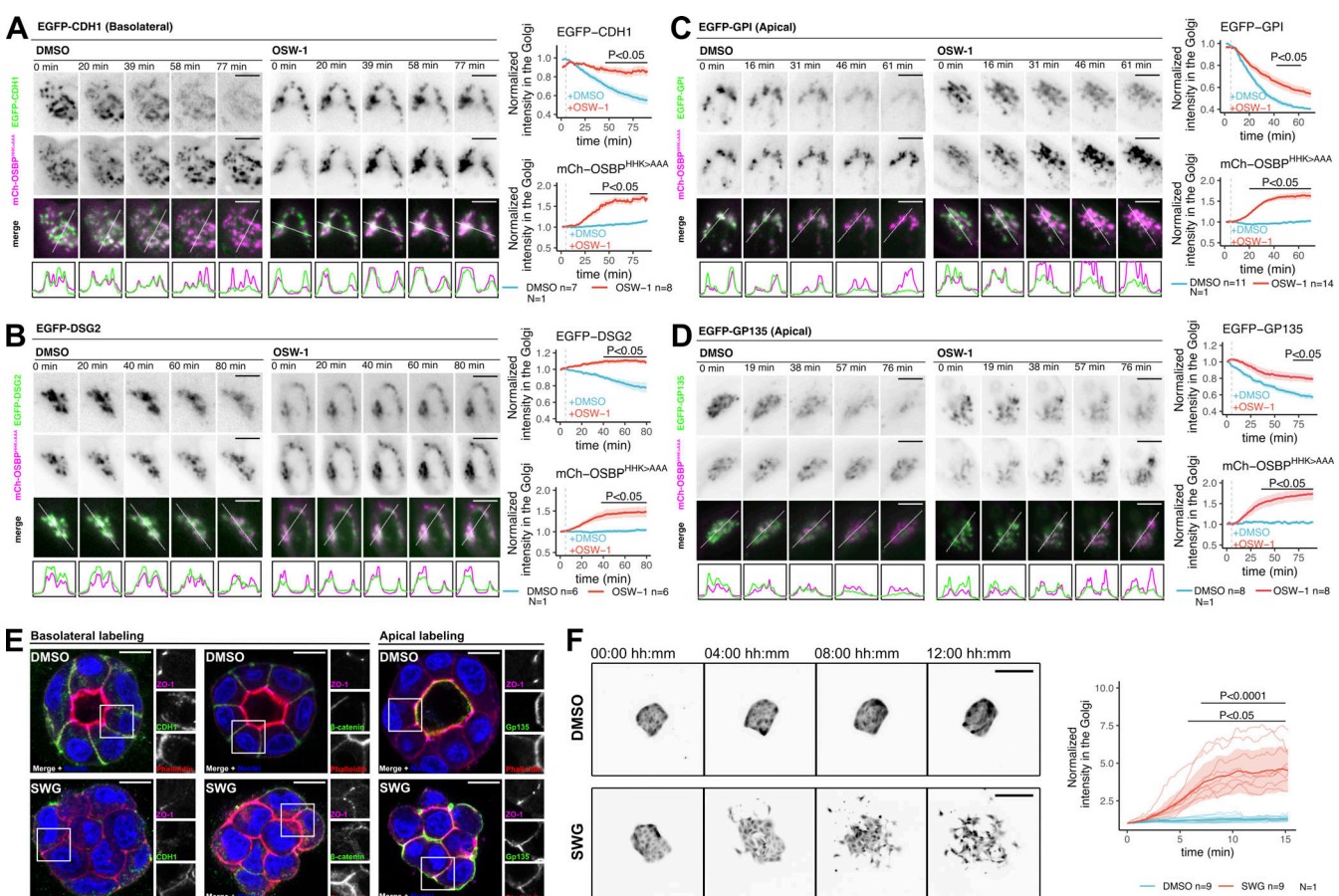

Figure 2. **OSBP inhibition affects apical and basolateral trafficking routes. (A–D)** MDCK cells were cotransfected with various RUSH model cargoes (CDH1- (A), DSG2- (B), EGFP-GPI- (C), GP135- (D)) and the OSBP MCS reporter mCherry-OSBP[HHK>AAA]. Cargo secretion was initiated by biotin addition. Cells were then kept at 19.5°C for 1.5 h to accumulate cargoes at the TGN. Following this, cells were transferred to 37°C to perform time-lapse live imaging. DMSO or OSW-1 was added to the cells 5 min following the start of the imaging. Intensity values of Golgi areas corresponding to both EGFP and mCherry channels were quantified and plotted. Bold lines with the shaded areas indicate mean ± SEM and indicated P values were calculated by two-way ANOVA followed by Sidak's multiple comparison test. N = number of independent experiments presented, n = number of cells used to calculate mean and error. The three independent experiments are shown in Fig. S2, A–D. Line plots indicate fluorescence intensities of each corresponding cargo and mCherry-OSBP[HHK>AAA] at the indicated areas. Bar = 10 µm. **(E)** MDCK cells were seeded into Matrigel to form cysts for 72 h. Cysts were then treated with SWG overnight. Representative cysts stained for apical/basolateral markers are shown. More cysts are shown in Fig. S2 E. Bar = 10 µm, diameter of the insets = 14.5 µm. **(F)** MDCK cells stably expressing EGFP were seeded into 12 well plates in low density and left to form cell colonies for 72 h. Following the addition of DMSO or SWG, colonies were analyzed by time-lapse fluorescence imaging. Cell scattering was quantified by measuring the colony perimeters. Each line corresponds to one colony and bold lines with the shaded area indicate mean ± 95% confidence interval. N = number of independent experiments presented, n = number of colonies analyzed. P values were calculated by two-way ANOVA followed by Sidak's multiple comparison test. Bar = 150 µm.

effect of these drugs on OSBP localization and lipid exchange activity. When OSBP is soluble in the cytoplasm, a high number of cytoplasmic proteins should be biotinylated. When OSBP is recruited to the MCSs, both ER and TGN-localized proximity proteins can be screened because the TurboID tag is appended to the OSBP C-terminal region, which transiently interacts with the two organelles. Due to its preference for disordered membranes, OSBP slides toward PI(4)P-rich regions along the TGN upon lipid exchange (Mesmin et al., 2017). When ORPphilins inhibit the lipid-transfer activity of OSBP, cholesterol levels drop and PI(4)P levels rise in the TGN membranes, which then provides new anchoring points for OSBP. This forced recruitment of OSBP leads to the formation of crowded, non-dynamic MCSs along the ER-TGN interface (Fig. 3 A; Burgett et al., 2011; Mesmin et al., 2017; Péresse et al., 2020). In good agreement

with this, in MDCK cells stably expressing OSBP-TurboID, the fusion bait was detected both in the cytoplasm and in βGalT1-positive Golgi structures (Fig. 3 B). Upon treatment with OSW-1 or SWG for 60 min prior to biotinylation, OSBP-TurboID showed increased recruitment to TGN membranes at the expense of the soluble form (Fig. 3, B and C). Remarkably, incubating the OSBP-TurboID-expressing cells for 2 h in a biotin-free medium after a 10 min biotin exposure led to the appearance of the biotinylation signal at the cell surface, whereas no such signal was observed when the cells were fixed directly after the 10 min biotin treatment (Fig. S3 B). This observation provided an additional hint that a large proportion of cell surface proteins visit OSBP-containing MCSs during their trafficking.

For proteomic analysis, we exposed MDCK cells stably expressing OSBP-TurboID to either DMSO, SWG, or OSW-1 for

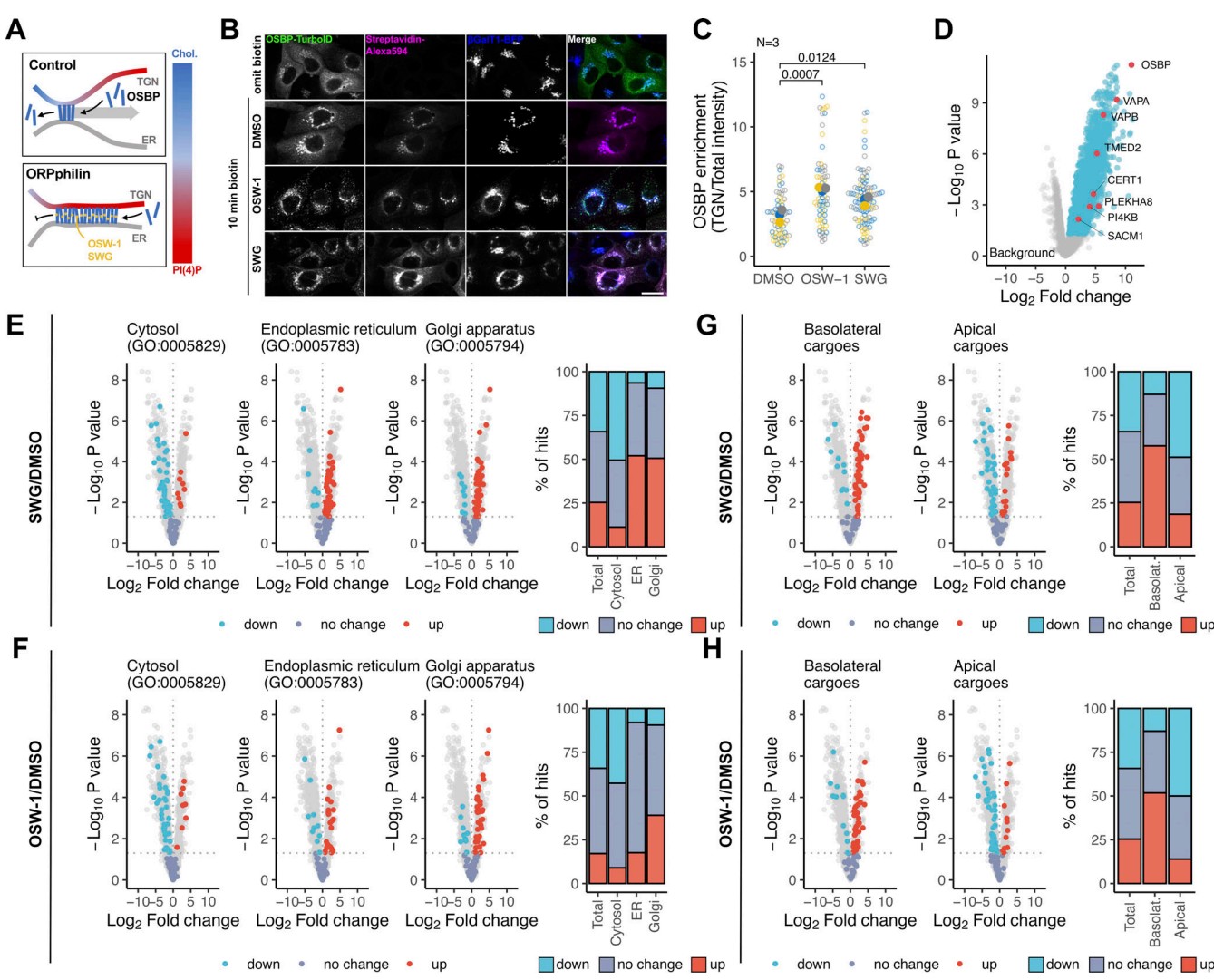

Figure 3. **OSBP proximity proteome reveals apicobasal cargo sorting in the TGN. (A)** OSBP cycles between its membrane-bound and cytosolic form. As a consequence of its lipid-transfer activity and its preference toward disordered membranes, OSBP contact sites slide toward PI(4)P-rich regions. When OSBP activity is inhibited by ORPphilins, its lateral dynamics and cycling between the cytoplasm and the contact sites are blocked. **(B and C)** MDCK cells stably expressing OSBP-TurboID were transfected with βGalT1-BFP to label TGN. Cells were treated with ORPphilins for 1 h and biotin was added to the cells in the last 10 min of the treatment when indicated. Cells were fixed, immunostained, labeled with Streptavidin-Alexa594 to visualize OSBP-TurboID and biotinylation, and then analyzed by confocal microscopy (B). Superplot showing TGN-enrichment of OSBP-TurboID upon OSW-1 and SWG treatments (C). N = number of independent experiments used to calculate means (filled circles). Means were compared by one-way ANOVA with Dunnett's post hoc test. Bar = 15 μm. **(D)** Hits that showed a significant increase in abundance values in TurboID-expressing MDCK over wild-type MDCK cells were considered members of the OSBP proximity proteome. Multiple proteins already described as partners or interactors of OSBP were identified. **(E and F)** Log$_2$-transformed fold change values obtained for SWG/DMSO (E) and OSW-1/DMSO (F) were plotted on volcano plots and proteins belonging to selected GO groups (Cytosolic proteins, ER-, and Golgi-resident proteins) were highlighted. Distribution of total hits and each group among the three statistical categories (no change, significantly more abundant-"up," significantly less abundant-"down") were plotted upon SWG (E) and OSW-1 (F) treatments. **(G and H)** Cargoes with highly basolateral or apical localizations were identified among the OSBP proximity partners and highlighted in the OSBP proximity landscape upon SWG (G) and OSW-1 (H) treatments. Upon ORPphilin treatments, basolateral cargoes become more abundant in the OSBP proximity.

60 min followed by biotin addition for the last 10 min of the ORPphilin treatments. We verified successful protein biotinylation and purification by streptavidin-HRP blotting and silver staining of the elution fractions (Fig. S3, C and D). Following on-bead trypsinization, peptides were subjected to mass spectrometry and proteins were identified by subsequent proteomics analyses.

Overall, we identified 1,507 proteins as members of the OSBP proximity landscape. In the DMSO-treated cells, we pinpointed many proteins already known to interact or co-localize with OSBP at MCSs. These include both VAP isoforms (VAPA and VAPB), the PI(4)P generating kinase PI4KIIIβ (PI4KB), and the ER-resident phosphatase SAC1 (SACM1L). Additionally, TMED2, the ceramide transporter CERT1 and the glucosylceramide-transfer protein FAPP2 (PLEKHA8) were also detected, confirming their collaboration with OSBP at the ER-TGN contact sites (Fig. 3 D; Anwar et al., 2022; Kumagai and Hanada, 2019; Mesmin et al., 2019).

To functionally annotate the OSBP proximity proteome, we performed a statistical enrichment analysis in cellular component GO terms. As expected, statistically significant enrichments were found in multiple GO groups encompassing ER and Golgi-resident proteins. However, other GO terms such as "plasma membrane," "cell junction," and "anchoring junction" showed significant enrichments as well, suggesting potential cargo clients of OSBP (Fig. S3 E). Besides the numerous adherent junction components (e.g., cadherins and catenins), we identified tight junction proteins (e.g., TJP1 and 2, Occludin, MarvelD2, and Claudin), integrins, CD44, EpCam, and the apical marker proteins PODXL (Gp135) and MUC1, all of which are regulators or determinants of epithelial polarity (Table S1).

We aimed to see whether cargoes with apical or basolateral localization are preferentially enriched in the OSBP proximity. For this, we correlated the abundances of the identified surface proteins among the proximity partners (expressed as -log-transformed P value over the background) with their apicobasal distribution (polarity value). These polarity values were determined by Caceres et al. (2019), who selectively labeled and quantified the apical and basolateral surface proteome of polarized MDCK cells using mass spectrometry. Strikingly, we observed no correlation, indicating that apical and basolateral cargoes are equally detected in the vicinity of active OSBP (Fig. S3 F).

Next, we assessed the effect of the OSBP inhibitors ORPphilins on the proximity proteome of OSBP (Fig. 3, E and F). For this, we divided the hits into three categories according to their fold change upon SWG or OSW-1 treatments over the DMSO-treated control: hits that showed no significant change, hits with increased (up), and hits with decreased (down) abundances. Then, we mapped specific classes of hits to the volcano plots and calculated their distribution along these three categories. Among all detected hits (total), ~25% of the proteins became more abundant (up) around OSBP upon SWG treatment, ~30% became less abundant (down), and about 45% of them showed no significant change (Fig. 3 E). Similar changes were detected after OSW-1 treatment (Fig. 3 F). As expected, the majority of ER and Golgi-resident proteins showed increased abundance around OSBP after SWG treatment, while most cytosolic proteins became less abundant (Fig. 3, E and F). These variations were in good agreement with imaging data (Fig. 3 B), showing that the proximity proteome of OSBP changes together with its subcellular localization.

Finally, we assessed the effect of ORPphilin treatments on secretory cargo abundance in the OSBP proximity landscape. For this, we once again used the Caceres dataset to assign apicobasal-distribution values to the identified cargoes in the proximity of OSBP (Caceres et al., 2019). First, we ranked the hits according to their polarized localization across the polarized MDCK plasma membrane. Thereafter, we selected extreme proteins (proteins with either high apical or basolateral localization) and we mapped these polarized surface proteins on the OSBP proximity landscape (Fig. S3 G; and Fig. 3, G and H). Strikingly, the abundance of most of the basolateral cargoes increased in the OSBP proximity upon ORPphilin treatments, whereas the majority of the apical cargoes became less abundant (Fig. 3, G and

H). This suggests that when OSBP contact sites extend but are deficient in lipid exchange due to ORPphilin treatments, they prefer to populate the TGN regions that favor the sorting of basolateral cargo proteins.

Overall, the proximity ligation strategy confirmed that endogenous apical and basolateral cargoes become close to OSBP during their Golgi transit. In addition, it suggested that apical and basolateral cargoes are segregated from each other in a manner that correlates with the positioning of OSBP-dependent MCSs on TGN membranes.

## OSBP inhibition perturbs surface expression of polarity determinants

In the RUSH assays, we found that the Golgi exit of apical cargoes is less susceptible to OSBP inhibitors than that of the basolateral cargoes. The proximity proteome confirmed that both apical and basolateral trafficking routes approach OSBP, and through the impact of ORPphilin treatments, it suggested that apical and basolateral cargoes are sorted across the lipid gradient generated by OSBP. However, the TurboID assay cannot report on the subsequent fate of the identified endogenous cargoes upon ORPphilin treatments. To directly address which cell surface proteins are perturbed by OSBP inhibition, we conducted a second proteomic analysis, now focusing on the dynamics of the MDCK surface proteome upon OSBP targeting.

We monitored the dynamics of the MDCK surface proteome after 0, 6, and 12 h SWG treatments using quantitative mass spectrometry (Fig. S3 H). Compared with time 0, we observed that 17 and 32% of the total detected surface proteome showed significant abundance changes upon 6 and 12 h SWG treatments, respectively (Fig. S3 H and Table S2). To define groups of surface proteins responding similarly to SWG treatment, we gathered the proteins that exhibited similar changes over the time course of the experiment using k-means clustering (Fig. 4 A). Cluster 1 gathers proteins showing a rapid decrease in abundance at the cell surface during the time course of the SWG treatment. Cluster 2 corresponds to proteins showing a gradual increase. Cluster 3 gathers proteins showing a slow decrease in surface expression upon SWG treatment. Last, proteins that showed a transient increase at the cell surface were grouped in Cluster 4 (Fig. 4 B and Table S2). Importantly, the majority of proteins were found in Clusters 1 and 3, indicating that SWG treatment downregulates the surface expression of a large number of proteins (Fig. 4 B).

We examined whether surface expression of apical and basolateral surface proteins show similar sensitivity to SWG treatment. For this, we performed an analysis similar to the one shown in Fig. 3, G and H. Using the Caceres dataset (Caceres et al., 2019), we identified highly apical and basolateral surface proteins and calculated their distribution across the four clusters. Strikingly, most of the basolateral surface proteins were gathered in Cluster 1, whereas the majority of the apical proteins were grouped in Cluster 3 (Fig. 4 C).

To confirm that basolateral cargoes are arrested in the TGN upon OSBP blockage, we localized endogenous E-cadherin in control and ORPphilin-treated MDCK cells using immunofluorescence (Fig. 4 D). As expected, E-cadherin disappeared from

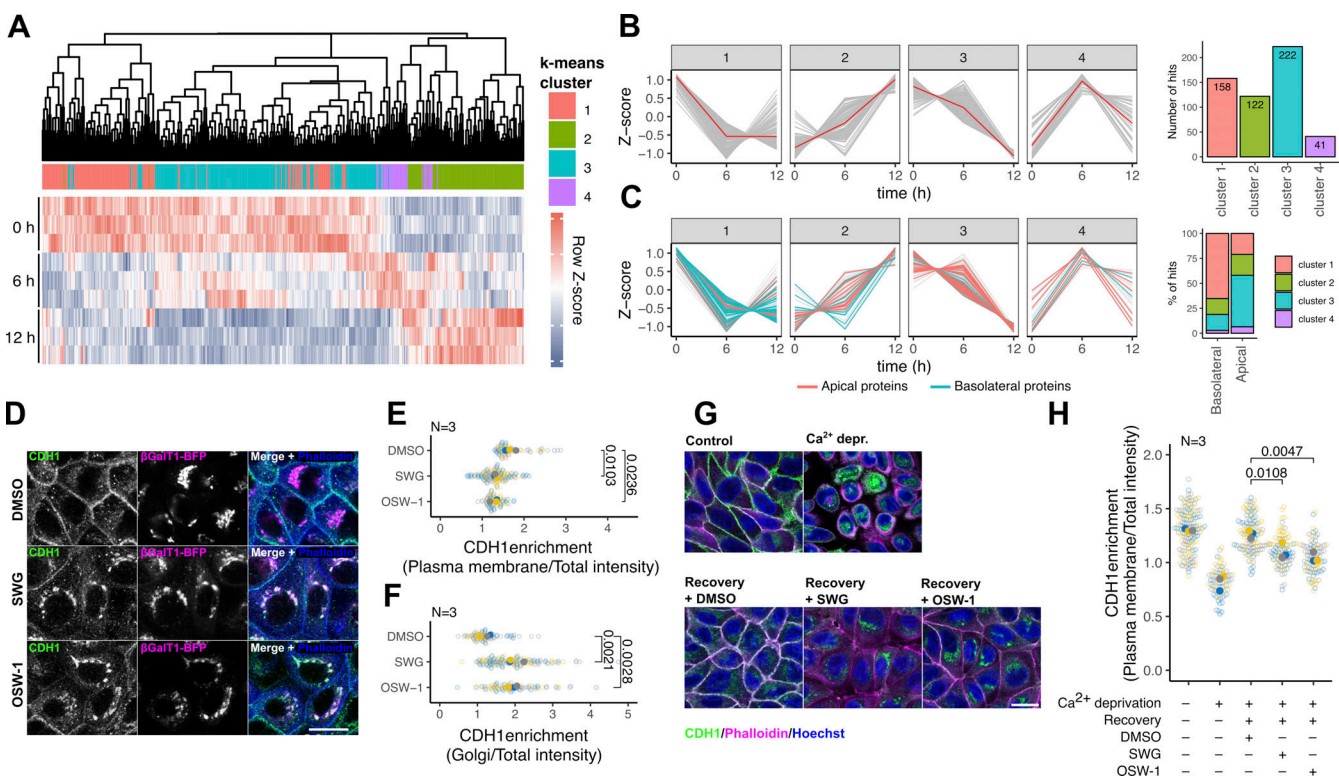

**Figure 4. OSBP targeting affects surface proteome dynamics. (A)** MDCK cells were treated with SWG for 0, 6, and 12 h. Surface proteins were then labeled and subsequently analyzed using quantitative mass spectrometry. Heat map showing the temporal dynamics of proteins that displayed a significant change in abundance on the cell surface at least at one time point compared to time 0. 4-color coding indicates k-means clusters. **(B)** K-means clusters show the temporal dynamics of each cluster. Red lines indicate cluster centroids. The bar plot indicates the distribution of the significantly changed surface proteins across the four k-means clusters. **(C)** Surface proteins with highly basolateral and apical localizations were highlighted among the SWG-perturbed proteins. The distribution of these hits indicates that most of the highly basolateral cargoes are grouped in cluster 1, whereas the majority of the apical proteins were gathered in cluster 3. **(D–F)** MDCK cells stability expressing βGalT1-BFP were treated with ORPphilins for 4 h and endogenous E-cadherin was immunolabeled and observed by confocal microscopy (D). Superplots showing E-cadherin enrichments in the plasma membrane (E) and in the TGN (F) upon ORPphilin treatments. $N$ = number of independent experiments to calculate means (filled circles). Means were compared by one-way ANOVA with Dunnett's post hoc test. Bar = 20 µm. **(G and H)** Confluent MDCK cells were kept in a calcium-free medium overnight and then placed in a regular culture medium in the absence or presence of ORPphilins. Cells were allowed to recover cell–cell junctions for 2 h, fixed, immunolabeled for endogenous E-cadherin and then imaged by confocal microscopy. $N$ = number of independent experiments to calculate means (filled circles). Means were compared by one-way ANOVA with Dunnett's post hoc test. Bar = 20 µm.

the surface after 4 h of ORPphilin treatments, and it concomitantly accumulated in intracellular structures positive for the Golgi resident βGalT1 (Fig. 4, E and F).

It has been established that epithelial cadherins have a very short half-life and that their surface-localized pool undergoes full renewal within hours (Brüser and Bogdan, 2017; Bryant and Stow, 2004; Cavey et al., 2008; McCrea and Gumbiner, 1991). When cultured in Ca$^{2+}$-free conditions, epithelial cells internalize and subsequently degrade cadherin molecules in a proteasome- and lysosome-dependent manner (Yamada et al., 2005; Shen et al., 2008). However, when Ca$^{2+}$ concentration is re-established, neosynthesized E-cadherin forms stable cell–cell junctions within a few hours. In our assay, control MDCK cells were able to recover E-cadherin-based cell–cell junctions 2 h following overnight Ca$^{2+}$ deprivation, whereas E-cadherin accumulated in the Golgi and failed to reach the cell surface in cells treated with ORPphilins during recovery (Fig. 4, G and H).

We concluded that the trafficking of basolateral proteins is more susceptible to OSBP blockage, although surface expression of both apical and basolateral proteins is perturbed by OSBP targeting.

## OSBP regulates polarized lipid distribution

At the molecular level, OSBP is not involved in direct interactions with vesicular trafficking machinery; instead, it directs cholesterol-PI(4)P lipid exchange at ER-TGN contact sites. In hTERT-RPE1 cells, this activity is massive and not only conditions the local distribution of cholesterol and PI(4)P but also determines the gradient of lipid order along the secretory pathway, including the lipid composition of the plasma membrane (Mesmin et al., 2017).

To investigate how this effect applies to the lipid composition of the polarized cell membrane domains, we assessed the effect of OSBP manipulations in polarized MDCK cells. First, we directly analyzed the distribution of cholesterol using cells stably expressing the cholesterol biosensor D4H-EGFP. In control siRNA conditions, the probe D4H-EGFP decorated the cell membranes of the polarized cells and an enrichment of the

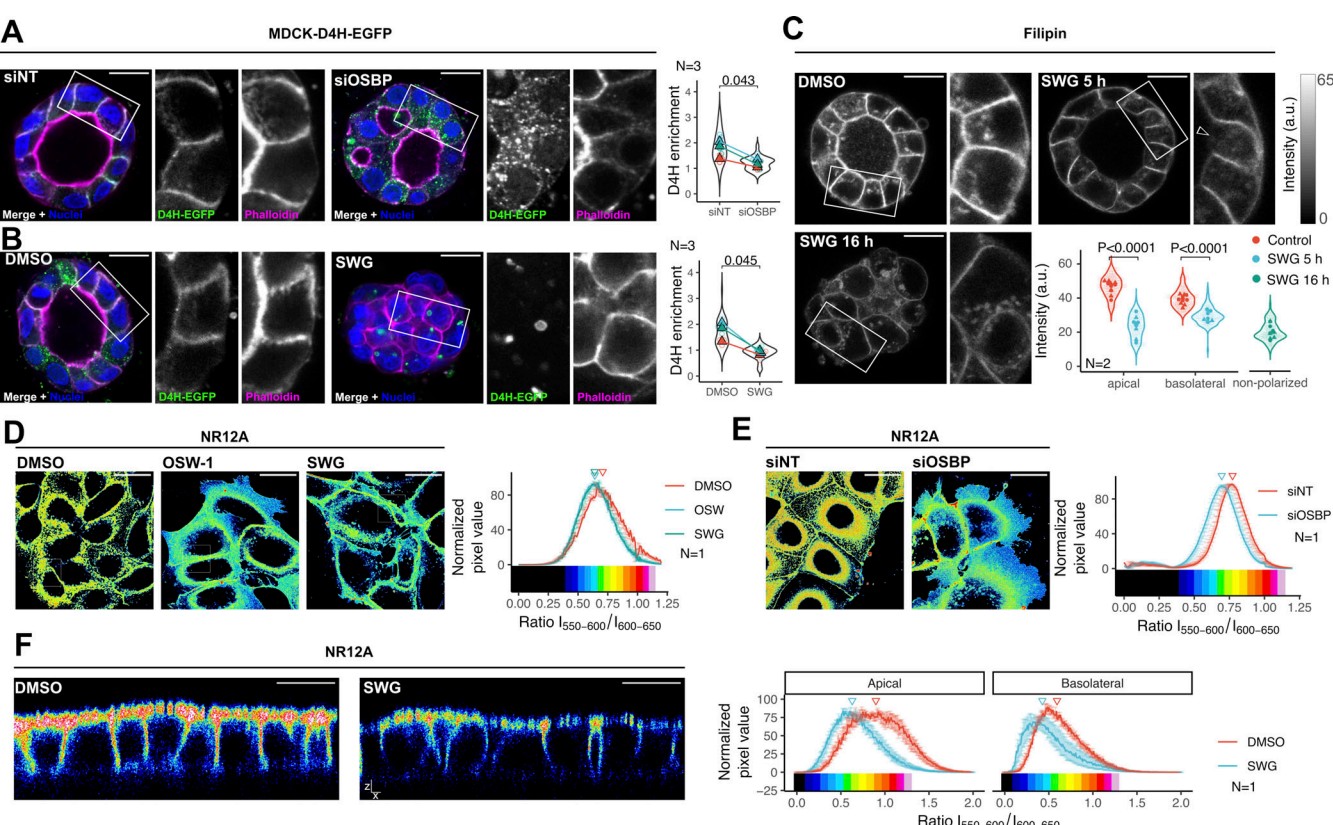

Figure 5. **OSBP regulates polarized cholesterol distribution. (A)** MDCK cells stably expressing D4H-EGFP were transfected with control or OSBP-silencing siRNA and reseeded into Matrigel 24 h later. Cysts were left to develop for 72 h, fixed, and imaged by confocal microscopy. D4H-EGFP signal on the cell surface significantly dropped upon OSBP silencing. Open circles on the superplot correspond to the mean intensity of one cyst, while filled triangles show the mean of all the analyzed cysts from each independent experiment. Violin plots indicate the distribution of single-cell intensity values across all experiments. N = number of independent experiments presented. Mean cyst intensities were compared by paired t test. More cysts are presented in Fig. S4 A. Bar = 15 µm, the diameter of the insets = 30.8 µm. **(B)** MDCK cells stably expressing D4H-EGFP were seeded into Matrigel. 72 h later, cysts were treated overnight with SWG. Following treatment, cysts were fixed and prepared for confocal imaging. D4H-EGFP signal on the cell surface significantly dropped upon OSBP inhibition. Open circles on the superplot correspond to the mean intensity of one cyst, while filled triangles show the mean of all the analyzed cysts in each independent experiment. Violin plots indicate the distribution of single cell intensity values across all experiments. N = number of independent experiments presented. Mean cyst intensities were compared by paired t test. More cysts are presented in Fig. S4 B. Bar = 15 µm, diameter of the insets = 30.8 µm. **(C)** MDCK cells were seeded in Matrigel, left to form cysts for 72 h, and then treated with SWG for indicated times. Cysts were fixed, extracted from Matrigel, and stained with Filipin to quantify plasma membrane cholesterol levels by confocal microscopy. Filled symbols on the superplot correspond to the mean intensity of one cyst, while open circles indicate single-cell intensities. N = number of independent experiments presented. Symbols correspond to cyst data obtained from independent experiments. Mean cyst intensities were compared by two-way ANOVA followed by Sidak's multiple comparison test. Arrowhead points to apical surface. Bar = 15 µm, the diameter of the insets = 30.3 µm. **(D and E)** 5 h ORPphilin-treated (D) or OSBP-silenced (E) MDCK cells were stained with the plasma membrane polarity dye NR12A, and confocal images were taken using emission windows corresponding to 550–600 and 600–650 nm. Ratio values are shown in the pseudocolored micrographs. N = number of independent experiments presented. Histograms show the mean ± SEM of four fields, each containing at least 8–10 cells. Triangles point to mean values. The three independent experiments are shown in Fig. S4, C–E and Fig. S4, F–H. Bar = 25 µm. **(F)** MDCK cells were filter-polarized and then treated with SWG for 5 h. Apical and basolateral surfaces were stained with the solvatochromic dye NR12A, then confocal images were taken using emission windows corresponding to 550–600 and 600–650 nm. The obtained ratiometric images were analyzed to obtain intensity histograms corresponding to apical and basolateral cell surfaces. N = number of independent experiments presented. Histograms show the mean ± SEM of five fields, each containing at least 8–10 cells. Triangles point to mean values. The three independent experiments are shown in Fig. S4, I–K. Bar = 10 µm.

fluorescent signal was detected at the basolateral domain of the plasma membrane (Fig. 5 A and Fig. S4 A). Upon OSBP silencing, a fraction of cysts exhibited morphological defects (see Fig. 1). Importantly, those cysts also showed altered cholesterol distribution: the D4H-EGFP cholesterol probe decreased at the plasma membrane and appeared in small, cytoplasmic structures (Fig. 5 A and Fig. S4 A). Similarly, the probe disappeared from the plasma membrane of MDCK cysts upon overnight treatment with SWG. Compared with OSBP silencing, the drug treatment exhibited a more potent effect (Fig. 5 B and Fig. S4 B).

Cholesterol is enriched and complexes with sphingomyelin at the apical surface of polarized epithelial cells (Gerl et al., 2012). However, due to its restricted affinity for free, accessible cholesterol pools, D4H does not reliably report apically localized cholesterol (Maekawa and Fairn, 2015; Maekawa, 2017). To test whether OSBP supplies cholesterol toward both apical and basolateral membranes or whether it is selective to one of these domains, we used filipin staining to visualize plasma membrane cholesterol in MDCK cysts treated with SWG at various times. In control cysts, we measured higher fluorescent intensities in the

apical membrane domains than in the basolateral segments, confirming the polarized distribution of cholesterol (Fig. 5 C). Remarkably, after 5 h of OSBP inhibition, conditions under which the cyst structure was still visible, significant drops in cholesterol levels in both apical and basolateral plasma membrane regions were observed (Fig. 5 C). After 16 h of SWG exposition, the cysts lost cell polarity and we observed a further drop in cell-surface cholesterol levels (Fig. 5 C).

We also used the plasma membrane-selective and solvatochromic polarity probe NR12A (Danylchuk et al., 2021) to detect changes in membrane lipid order upon OSBP inhibition/silencing. In non-polarized cells, we observed that ORPphilin treatments (Fig. 5 D and Fig. S4, C–E) or OSBP silencing (Fig. 5 E and Fig. S4, F–H) decreased the ratio values of NR12A, indicating that OSBP targeting leads to a less ordered plasma membrane. On filter-polarized MDCK cells, we observed that ratio values corresponding to both apical and basolateral domains shifted toward smaller values following 5 h SWG treatment, again indicating decreased lipid order in both domains of the polarized plasma membrane (Fig. 5 F and Fig. S4, I–K).

Altogether, these various imaging approaches indicate that OSBP supplies both apical and basolateral domains with cholesterol. This wide effect on cholesterol distribution is in good agreement with the effect of OSBP inhibition on both the apical and basolateral trafficking routes as determined before.

**Epithelial-to-mesenchymal transition targets OSBP**
Because OSBP regulates epithelial sorting and subsequent trafficking of polarized cargoes that determine apicobasal polarity, we aimed to test whether this feature of OSBP is general across many cell types or restricted to epithelial cells.

First, we determined whether the biochemical activity of OSBP changes upon epithelial-to-mesenchymal transition, i.e., when epithelial cells are losing their apicobasal polarity. For this, we induced EMT in MDCK cells using HGF and in A549 adenocarcinoma cells using TGF-β (Kubiczkova et al., 2012; Hao et al., 2019; Hua et al., 2019). Thereafter, we measured PI(4)P levels at the Golgi area upon OSBP inhibition. By inhibiting OSBP, OSW-1 protects PI(4)P from the OSBP cycle, thereby allowing the estimation of the PI(4)P pool that is consumed specifically by OSBP in the cell (Mesmin et al., 2017). Following OSW-1 addition, Golgi-localized PI(4)P levels increased rapidly and reached a plateau within 10–15 min in both MDCK and A549 cells (Fig. 6, A and B; and Fig. S5, A and B). However, the difference between the steady state and maximal values was larger in control cells than in HGF or TGF-β-induced mesenchymal cells (Fig. 6, A and B; and Fig. S5, A and B). Quantification of the PI(4)P signal before and after OSW-1 addition suggested that OSBP consumed ∼95% of the cellular PI(4)P pool in control MDCK cells compared with only ∼60% in HGF-treated cells. In A549 cells, OSBP consumed ∼75% of the total PI(4)P pool at steady state and only 40% in TGF-β-induced cells. These observations indicated that EMT correlates with a significant reduction of OSBP activity (Fig. 6, A and B). Thus, OSBP is biochemically more active in epithelial cells than in mesenchymal cells.

To test whether other members of the ORP family contribute to epithelial and mesenchymal phenotypes, we performed a bioinformatics screen to correlate the expression level of OSBP and the other members of the ORP family across cell lines with different epithelial characteristics (Rajapakse et al., 2018). We extracted the gene expression data of the cell lines included in the Cancer Cell Line Encyclopedia (Barretina et al., 2012), focusing on all ORP genes as well as a panel of epithelial and mesenchymal marker genes (Fig. 6 C). To define a cell-line specific epithelial–mesenchymal signature, we calculated a single epithelial–mesenchymal index (EMI) for each cell line using the marker gene expression data (Fig. 6 D) and we correlated these values with the expression levels of the various ORPs (Fig. S5 C and Fig. 6 E). This analysis indicates that OSBPL2 and OSBP expression levels correlate positively with EMI, while OSBPL8 and OSBPL6 expressions show a negative correlation. Thus, OSBPL2 and OSBP are highly expressed in cell lines with epithelial characteristics, while OSBPL8 and OSBPL6 are rather mesenchymal cell-specific genes.

To experimentally investigate the differential ORP gene expression upon EMT, we triggered EMT by TGF-β treatment in A549 lung adenocarcinoma cells; then, we assessed the relative transcript levels of EMT transcription factors and the top-changed ORPs by RT-qPCR. While snai1- and snai2-inductions dominated the TGF-β-triggered EMT in A549 cells, we observed a concomitant decrease in OSBP and OSBPL2 transcript levels. A significant increase in OSBPL6 messengers was measured, validating our previous findings, whereas no change in OSBPL8 expression was observed (Fig. 6 F).

To test whether this observation can be extended to disease conditions as well, we repeated the analysis using RNAseq data obtained from TCGA lung cancer specimens (Fig. S5, D–G). In agreement with the cell-line collection data, OSBP and OSBPL2 expressions positively correlate with the epithelial features of the given cancer tissue specimens.

To further investigate the epithelial expression of OSBP and OSBPL2, we assessed histology samples of the Human Protein Atlas (Uhlen et al., 2015). As revealed by specific antibodies, OSBP and OSBPL2 proteins exclusively decorate the epithelial cell layers of various tissue samples, such as nasopharynx and urinary bladder, while the skeletal muscle, which has a mesenchymal origin, lacks OSBP and expresses high levels of OSBPL6 (Fig. S5 H).

Loss of epithelial polarization through epithelial-to-mesenchymal transition (EMT) is frequently observed in various stages of malignancies. EMT can lead to metastatic dissemination; therefore, tumors with mesenchymal characteristics are typically coupled to shorter patient survival (Dongre and Weinberg, 2019; Aruga et al., 2018). By analyzing lung adenocarcinoma (LUAD) survival data, we found that high OSBP and OSBPL2 expressions are associated with significantly longer patient survivals, suggesting that downregulation of these genes is a signature of tumor malignancy (Fig. 6, G and H).

Altogether, these analyses suggest that (i) OSBP is highly expressed in epithelial cells, (ii) it is targeted by EMT, and (iii) it might have an impact on the overall outcome of some malignancies.

## Discussion
It has long been shown that the TGN is a major cargo sorting station for the formation of transport intermediates delivered to

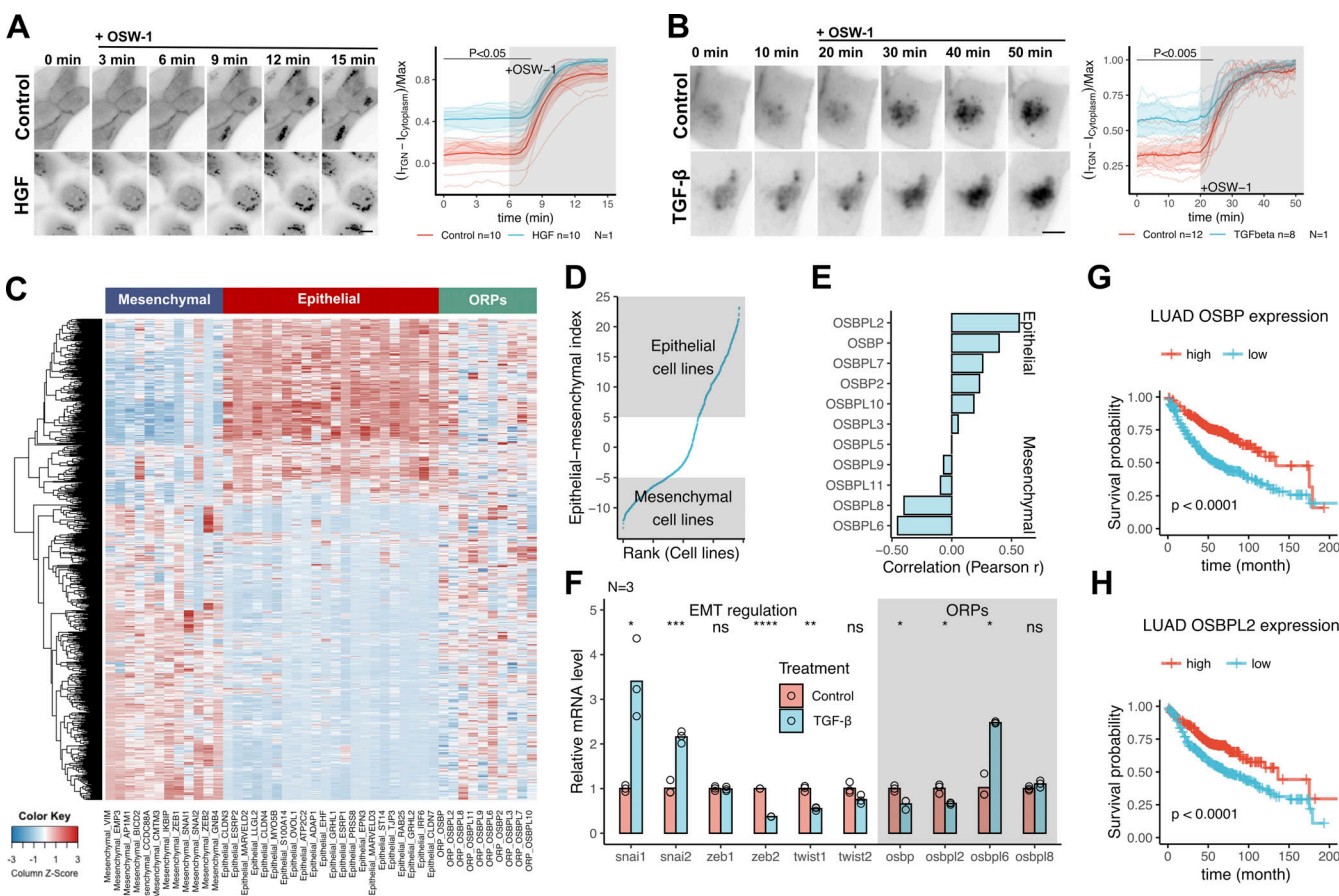

Figure 6. **OSBP is highly expressed in epithelial cells. (A and B)** EMT was induced in MDCK cells stably expressing P4M-SidM-EGFP by HGF treatment (A). TGF-β was used to induce EMT in A549 cells (B), and then cells were transfected with P4M-SidM-EGFP plasmid. Cells were mounted for epifluorescent live cell imaging to asses OSBP-dependent PI(4)P consumption. OSW-1 was added to the cells at the indicated time points after the start of imaging. Intensities in Golgi areas were measured to plot the kinetics of the P4M-SidM-EGFP experiments. N = number of independent experiments presented, n = number of cells analyzed. Each line corresponds to one cell and bold lines with the shaded area indicate mean ± 95% confidence interval. P values were calculated by two-way ANOVA followed by Sidak's multiple comparison test. The three independent experiments are shown in Fig. S5, A and B. Bar = 10 μm. **(C)** Expression heatmap of epithelial markers, mesenchymal markers and ORP genes across cell lines of the Cancer Cell Line Encyclopedia collection. **(D)** Using the marker gene expression data, epithelial-mesenchymal indexes were calculated for each cell line and ranked. **(E)** Epithelial-mesenchymal indexes were correlated with ORP expressions in each cell line then Pearson r values were ranked to reveal ORPs associated with epithelial or mesenchymal cells. OSBPL2 and OSBP are epithelial, while OSBPL6 and OSBPL8 are mesenchymal-specific ORPs. **(F)** EMT was triggered in A549 cells with 72 h TGF-β treatment, then RT-qPCR was performed to assess expression changes in EMT-transcription factors and selected ORPs. Means are calculated from three independent experiments. *P < 0.05; **P < 0.01; ****P < 0.001; ns = non-significant, Welch's t test. **(G and H)** OSBP and OSBPL2 expression data was obtained from TCGA Lung adenocarcinoma samples (n = 719) and patients were split according to median gene expression values to obtain high and low expression groups. Low OSBP and OSBPL2 expressions are associated with a poor clinical outcome in lung adenocarcinoma patients. P values were calculated by Gehan–Breslow–Wilcoxon test.

the apical or basolateral surface of epithelial cells. Furthermore, evidence suggested that such sorting relies not only on protein recognition mechanisms, typically coat/adaptor-mediated vesicle budding, but also on lipid-mediated mechanisms, possibly on cholesterol and sphingolipid enriched regions (Cao et al., 2012; Keller and Simons, 1997; Deng et al., 2016; Boncompain and Weigel, 2018; Lippincott-Schwartz and Phair, 2010). However, despite progress in imaging cargo and lipid sorting at the TGN, tools for modifying or isolating lipid membrane regions are limited, making it difficult to unravel the contribution of lipid-mediated mechanisms on cargo-selective secretion. Taking advantage of the identification of OSBP as a prominent transporter of cholesterol toward the TGN and of OSBP-specific pharmacological tools, we show here that OSBP is essential for proper

cargo sorting at the TGN and their subsequent transport in epithelial cells. The observation of cargo waves with the same periodicity as lipid waves arising from the OSBP/PI(4)P cycle suggests that OSBP acts by creating lipid gradients along the extensive tubular network of the TGN, thereby facilitating lateral sorting of cargo proteins.

OSBP burns more PI(4)P in epithelial than in mesenchymal cells. Cysts developed from OSBP-silenced cells display morphological defects. Furthermore, OSBP inhibition by the specific drug SWG leads to a complete loss of epithelial polarity, which is accompanied by reduced cholesterol levels in both the apical and basolateral regions. Altogether, these observations indicate a key function of the lipid exchange activity of OSBP in maintaining the epithelial phenotype.

Targeting OSBP leads to an altered distribution of cholesterol at the polarized plasma membrane; however, this effect is not restricted to lipids. We found that a large portion of the epithelial plasma membrane proteome is perturbed upon OSBP inhibition as well, highlighting that OSBP controls the trafficking of various surface-localized cargoes. Furthermore, by interrogating the cargoes in the proximity of OSBP during their trafficking, we revealed a major role of OSBP activity in post-Golgi trafficking of endogenous cargoes with a polarized destination. Strikingly, among the repertoire of the OSBP proximity partners, apical and basolateral cargoes were equally identified.

At first glance, the cholesterol transfer activity of OSBP should favor the exit of cargoes that prefer a cholesterol-rich membrane environment, such as apically sorted proteins. However, inhibition of OSBP prevented post-Golgi egress of basolateral cargoes more efficiently. This effect should be considered in light of a fundamental, albeit counterintuitive, property of OSBP: its membrane-binding properties somehow oppose its lipid exchange activity (Mesmin et al, 2017, 2019).

OSBP binds to TGN via its PH domain, which recognizes PI(4)P. OSBP has a dwell time on PI(4)P-containing liposomes of tens of minutes (Jamecna et al., 2019), suggesting that OSBP moves along the TGN membrane mainly by lateral diffusion until the PI(4)P level drops. Furthermore, the PH domain of OSBP has an extended loop that necessarily inserts into the lipid membrane, which is not compatible with tight lipid packing (Lenoir et al, 2010, 2015). Thus, OSBP binds specifically to PI(4)P-rich disordered membranes. However, during the lipid exchange activity of OSBP, cholesterol gets enriched and PI(4)P drops. This conflict between the lipid binding properties of OSBP and its lipid exchange activity creates a negative feedback loop that promotes the displacement of OSBP-contact sites along the TGN and should create gradients between PI(4)P-rich/cholesterol-poor membrane regions and PI(4)P-poor/cholesterol-rich lipid regions along the Golgi membrane (Mesmin et al., 2017). Ultimately, these gradients should promote lateral segregation and clustering of protein cargoes according to their affinity for membrane regions with different lipid composition and packing properties (e.g., accumulation of apical proteins at cholesterol-rich regions). Notably, when OSBP lipid turnover is blocked by ORPphilin treatments, PI(4)P levels rise, which recruits more OSBP to the contact sites. Ultimately, this catalytically blocked and non-dynamic OSBP populates PI(4)P-rich TGN regions. Thus, the forced recruitment of OSBP to such PI(4)P-rich membranes physically blocks the trafficking of cargoes that favor these TGN regions. As post-Golgi trafficking of basolateral proteins was more sensitive to ORPphilin treatments and their abundance in the OSBP proximity increased under such conditions, we conclude that PI(4)P-rich TGN regions are sites of basolaterally destined cargoes. Conversely, apical proteins are sorted on cholesterol-rich TGN membranes (Fig. 7). Of note, no such cargo type-specific blockage was observed in OSBP-silenced cells since the lack of cholesterol/PI(4)P gradient led to sorting defects rather than a massive cargo type-specific trafficking arrest (Fig. 7).

Importantly, we observed transient oscillations of a model cargo as it traversed the Golgi, and these waves had the same

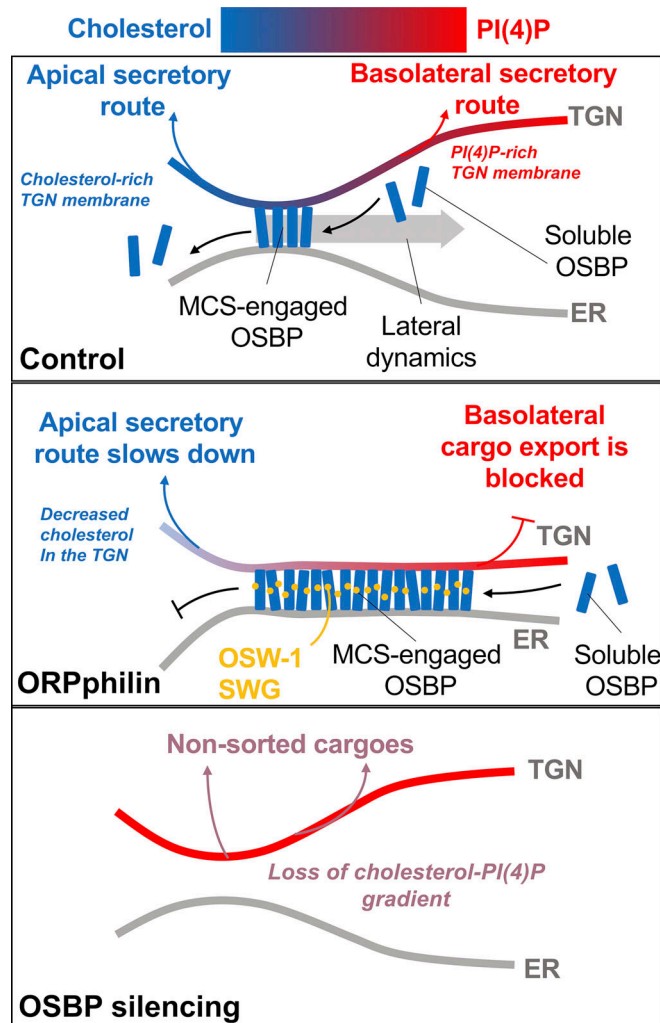

Figure 7. **Model for OSBP function in apicobasal cargo sorting.**

periodicity as those of PI(4)P. Moreover, MDCK cells showed pronounced OSBP/PI(4)P traveling waves in the Golgi under steady-state conditions, whereas this feature is less evident in other cell types (Mesmin et al., 2017). Pharmacological inhibition of PI4KIIIβ leads to the dissociation of OSBP from TGN and disrupts PI(4)P traveling waves. This inhibition impairs apicobasal cargo sorting and results in a striking loss of polarized morphology. Thus, the OSBP/PI(4)P axis is particularly developed in epithelial cells where it facilitates the formation of lipid gradients along the TGN for associated cargo sorting.

Clearly, such dynamic behavior presents strong experimental challenges for the future. To simultaneously observe the OSBP cycle, the distribution of associated lipids (not only PI(4)P but also cholesterol) and the fate of the various cargoes and their associated machinery in epithelia will require exquisite imaging capabilities. Nevertheless, our work gives a first indication of the importance of the OSBP cycle for a specific cellular phenotype. OSBP deletion in mice is lethal at early stages of embryogenesis (Brown and Goldstein, personal communication), whereas OSBP-targeting drugs have many potential therapeutic applications (Roberts et al., 2019; Bensen et al., 2021). Since the

formation of epithelial cell layers is indispensable during early development, understanding the OSBP-dependent cargo sorting mechanisms precisely is mandatory for the rational development of therapies targeting membrane contact sites.

## Materials and methods

### Plasmid generation, reagents, and pharmacology
Str-KDEL-SBP-EGFP_GP135 was a gift from S. Miserey (Institut Curie, Paris, France; Fourriere et al., 2019). To replace the EGFP-tag with mCherry in the Str-KDEL_SBP-EGFP-Ecadherin construct, EGFP was digested from the plasmid using SbfI and FspI restriction enzymes and then the mCherry cDNA amplified by sequence-specific primers was inserted into the plasmid by blunt ligation. To generate DSG2-expressing RUSH constructs, the E-cadherin containing RUSH vector was cut with XbaI and then the overhanging nucleotides were blunted by T4 polymerase. E-cadherin was cut from the linearized vector using FseI and the resulting Str-KDEL_SBP-EGFP plasmid was used for further cloning steps. Sequence-specific primers containing FseI restriction sites and blunt ends were applied on cDNA obtained from A549 cells. Following digestions, amplicons were ligated into Str-KDEL_SBP-EGFP plasmid backbones. To generate OSBP-TurboID-HA, we amplified OSBP cDNA using sequence-specific primers containing NheI and HindIII cutting sites from an OSBP-mCherry expressing plasmid described earlier (Mesmin et al., 2013) and then the amplicon was inserted into a pcDNA3.1-Zeo(+) vector. Using specific primers containing HindIII and BamHI restriction sites, TurboID was amplified from a reference template plasmid and the sequence was inserted into this vector to generate an OSBP-TurboID-HA expressing construct. All the generated plasmids were subjected to capillary sequencing prior to application to confirm that their sequences were correct. Primer sequences used for cloning are listed in Table S3.

OSW-1 was a generous gift from Matthew D. Shair (Harvard University, Cambridge, MA) while SWG was synthesized and provided by Fanny Roussi (Centre National de la Recherche Scientifique ICSN, France). For the proximity biotinylation assay, SWG was used at 0.5 µM and OSW-1 was applied at 50 nM concentrations for 1 h. Upon RUSH experiments, OSW-1 was applied at 20 nM, while SWG was used at 0.5 µM. SWG was used overnight on MDCK cysts at 0.5 µM concentration. For experiments shown Fig. 3, D–I, OSW-1 was applied at 2 nM, while SWG was used at 0.5 µM. To measure OSBP-dependent PI(4)P consumption in living cells, OSW-1 was applied at 50 nM on GFP-P4M-SidM-expressing cells. PIK93 was purchased from Sigma-Aldrich and used in 500 nM concentration. PI4KIIIbeta-IN-10 (MedChemExpress) was used at 100 nM for the experiments shown in Fig. S1, C and D, and at 25 nM for the experiments shown in Fig. S1, F and G.

Detailed information on the origin, application, and reference of plasmids and reagents used in this study are listed in Table S3.

### Cell culture, EMT, plasmid and siRNA transfection, 3D cyst formation, and calcium switch
MDCK cells were cultured in Minimal Essential Medium with Earl's balance salts (EMEM; Sigma-Aldrich) completed with 5% fetal bovine serum and 1% ZellShield (Minerva Biolabs). A549 cells were maintained in Dulbecco's Modified Eagle's Medium (DMEM; Gibco) complemented with 10% FBS and 1% ZellShield. hTERT-RPE1 cells were cultured in DMEM/F12 (Gibco) medium completed with 10% FBS and 1% ZellShield. Cells were maintained in a 37°C humidified incubator kept in a 5% $CO_2$ atmosphere.

To transfect plasmid DNA and siRNA (Thermo Fisher Scientific), MDCK cells were nucleofected by Amaxa Cell Line Nucleofector using Kit L (Lonza), while Kit V was used for hTERT-RPE1 nucleofections. Plasmid DNA was transfected into A549 cells by Xfect transfection reagent (Takara Bio). siRNA target sequences can be found in Table S3.

To induce EMT in A549 cells, $0.25 \times 10^6$ cells were seeded into 10-cm-diameter cell culture dishes and then treated with 5 ng/ml TGF-β (Abcam) on the next day. TGF-β was kept with the cells for at least 72 h. Cells were reseeded into appropriate culture dishes for experiments, maintaining a 5 ng/ml TGF-β concentration in the medium throughout the experiment.

To form MDCK cysts, 2,000 trypsin-individualized cells were resuspended in 20 µl Matrigel (Corning) and dropped on eight- or four-well µ-Slide chambered coverslips (Ibidi). Following the polymerization of the gel, a cell culture medium was placed on the samples and cysts were left to develop for 72 h.

Upon calcium switch experiments, confluent MDCK cultures were washed twice with PBS and then cells were incubated overnight in calcium-free medium (S-MEM; Gibco) supplemented with 5% dialyzed FBS. Following this, a regular medium with or without ORPphilins was applied to the cells for indicated times.

### Generation of stable cell lines
To generate stable cell lines, MDCK cells were transfected with linearized plasmid DNA and selected with either 500 µg/ml G418 (Sigma-Aldrich) or 400 µg/ml Zeocin (Gibco) 48 h following transfection for 2 wk. Surviving colonies were picked, amplified, and tested for transgene expression by immunoblotting or fluorescent microscopy. For the proximity labeling assays, clones expressing OSBP-TurboID-HA with a similar level to that of the endogenous OSBP were selected.

### Scattering assay and image analysis
For cell scattering assays, MDCK-EGFP cells were seeded onto 12-well plates at 2,000 cell/well density and then cell clusters were left to grow for 3 d. Following the addition of 100 nM SWG, time-lapse imaging was performed in a Cytation5 Cell imaging multimode-reader set to 37°C containing 5% $CO_2$ and operated by Gen5 software. Cluster scattering was quantified by measuring object perimeters over time-lapse imaging by CellProfiler 4.1.3 image analyses software.

### RT-qPCR
To analyze the mRNA expression of selected genes, $0.2 \times 10^6$ A549 cells were seeded into 6 cm–diameter culture dishes and then EMT was induced with 5 ng/ml TGF-β for 72 h. Total RNA was purified using the RNeasy Plus Mini kit (Qiagen) and a total amount of 2 µg RNA was reverse transcribed with a GO Script

Reverse Transcriptase kit (Promega) using random primers. Relative transcript levels were measured in a LightCycler 480 Instrument (Roche) using Takyion No Rox SYBR MasterMix (Eurogentec). Gene-specific primer sequences are summarized in Table S3. To calculate gene expression levels, the ΔΔCT method was applied using GAPDH as the reference gene.

## Immunofluorescence and confocal microscopy

For immunofluorescent labeling, cells were seeded onto eight- or four-well μ-Slide chambered coverslips (Ibidi). Following the experimental procedures, cells were fixed using 4% PFA and quenched in 50 mM $NH_4Cl$. Cells were permeabilized with 0.5% Saponin in 2% BSA-PBS and then antibodies were diluted in the same buffer and incubated with the cells overnight at 4°C. Following three washing steps with PBS, secondary antibodies were applied for 1 h together with Hoechst (Invitrogen) and other dyes such as DraQ5 (Cell Signaling Technology), Phalloidin (Invitrogen), and Streptavidin594 (Invitrogen) depending on the experiment. A detailed list of antibodies and other materials applied for immunolabelling is collected in Table S3.

Immunolabeled samples were analyzed with a Zeiss LSM 780 confocal microscope operated with ZEN software using a Plan-Apo- chromat 63X/1.4 Oil objective (Carl Zeiss). CellProfiler 4.1.3 image analysis software was applied to assess confocal micrographs by custom-made pipelines.

## Filipin staining and NR12A imaging

Following ORPphilin treatments, MDCK cysts were fixed in 1.5% PFA, extracted from the Matrigel using Cultex Organoid Harvesting Solution (R&D systems), and then collected by centrifugation. After two washing steps with PBS, cysts were stained with 50 μg/ml Filipin (Sigma-Aldrich) for 45 min in the dark. Following three washing steps, cysts were transferred into four well μ-Slide chambered coverslips (Ibidi) and visualized with a Zeiss LSM 780 confocal microscope operated with ZEN software using a Plan-Apo- chromat 63X/1.4 Oil objective (Carl Zeiss). Domain-selective fluorescent intensities were quantified on the obtained images using Fiji image processing software (Schindelin et al., 2012).

For solvatochromic imaging of plasma membrane lipid order, control, or siRNA-nucleofected MDCK, cells were seeded onto the four-well μ-Slide chambered coverslips (Ibidi). For filter polarization, cells were seeded onto the bottom surface of 0.4 μm-pore sized transwell filter units (Corning) and then left to form polarized cell layers for 5 d. Following 5 h exposition to 0.5 μM SWG, cells were washed three times with prewarmed PBS and 10 nM NR12A diluted in serum-free medium was added to the top and bottom chambers for 10 min at room temperature to label apical and basolateral surfaces, respectively. A drop of culture medium was applied on a 35 mm–diameter μ-Dish (Ibidi), and then transwells were mounted to perform confocal imaging using a Leica SP8 STED 3X (Leica Microsystems) equipped with a pulsed white laser as an excitation source. All images were acquired through a 93X/1.3 glycerol objective using the LAS X software (Leica Microsystems). The dye was excited at 488 nm wavelength and images were taken using emission windows corresponding to 550–600 and 600–650 nm wavelengths. Ratiometric images were generated using a specific Fiji macro by dividing pixel intensities corresponding to 550–600 nm images by their 600–650 nm counterparts and then the resulting images were transformed to a pseudocolor scale. Apical and basolateral membrane domains were selected manually and ratio value distributions were measured by Fiji to obtain histograms.

## Epifluorescence live cell imaging and image analysis

For live epifluorescence imaging experiments, nucleofected MDCK cells were seeded into 35-mm-diameter μ-Dishes (Ibidi) in $5 \times 10^5$ cell/dish density and incubated for 24 h before imaging. Control or TGF-β-stimulated A549 cells were seeded at $0.2 \times 10^6$. Prior to imaging, cells were washed three times with prewarmed PBS and the medium was replaced with 1 ml phenol red–free culture medium and samples were mounted in a 37°C stage chamber (Okolab). For RUSH experiments, the release of secretory model cargoes was induced by adding biotin to the cells at 80 μM together with DMSO or specific drugs at the indicated time points. Time-lapse imaging was carried out by an IX83 inverted microscope (Olympus) equipped with an iXon3 camera (Andor) and an UPlanSApo 60X/1.35 oil objective (Olympus) and operated by a MetaMorph software (Molecular Devices). Generated image sequences were analyzed by Fiji image processing software using the StackReg plugin.

## Surface and proximity labeling, proteomics analyses by NanoHPLC-HRMS

For the surface proteomics experiment, $1 \times 10^7$ MDCK cells were seeded into 150-mm cell culture dishes in triplicates/condition and left to settle overnight. On the following day, cells were treated with 100 nM SWG for the indicated times. Surface proteins were then labeled using EZ-Link Sulfo-NHS-SS-Biotin included in a Thermo Fisher Scientific Pierce Cell Surface Biotinylation and Isolation Kit (Thermo Fisher Scientific) following the instructions of the manufacturer. Following quenching with TBS, cells were lysed with the provided Lysis Buffer completed with cOmplete Protease Inhibitor Cocktail (Roche), and protein concentrations were determined using a BCA assay kit (Thermo Fisher Scientific).

For the OSBP proximity ligation assay, $1 \times 10^7$ wild-type MDCK or MDCK cells stably expressing OSBP-TurboID were seeded into 150-mm diameter dishes in triplicates/condition. The next day, cells were either treated with DMSO, 0.5 μM SWG, or 50 nM OSW-1. After 50 min, biotin was added to the culture mediums at 0.5 mM concentration and samples were incubated for an additional 10 min at 37°C. Cells were then washed five times with ice-cold PBS and lysed in RIPA buffer supplemented with cOmplete Protease Inhibitor Cocktail (Roche). Cell lysis was facilitated by sonication, and following centrifugation, protein concentrations were determined from the supernatant using a BCA assay kit (Thermo Fisher Scientific). Lysates were subjected to immunoblotting to confirm OSBP-TurboID expression and successful protein biotinylation (Fig. S3 D).

Sample preparation for mass spectrometry obtained from surface labeling or proximity ligation was carried out as

described elsewhere (Branon et al., 2018; Hung et al., 2016) with minor modifications. Briefly, 1.5 mg of total protein/sample from the surface labeling and 5 mg of total protein/sample from the proximity ligation were combined with 330 µl RIPA-equilibrated Pierce Streptavidin Magnetic Beads (Thermo Fisher Scientific) and incubated on a rotator at 4°C overnight. Beads were washed two times with RIPA buffer, then once with 1 M KCl, 0.1 M Na$_2$CO$_3$, once with 2 M urea diluted in 10 mM TRIS-HCl pH = 8.0, and twice with RIPA buffer again. Approximately, 5% of the slurry was saved to verify the successful enrichment of biotinylated material by silver staining (Fig. S3 D). Beads were further washed twice with 50 mM Tris-HCl pH = 7.5 and twice with 2 M urea diluted in 50 mM Tris-HCl pH = 7.5. Then on-bead digestion was performed by adding 0.4 µg Pierce MS grade trypsin protease (Thermo Fisher Scientific) and 1 mM DTT in 2 M urea/50 mM Tris-HCl pH = 7.5. Trypsinization was performed at 25°C with agitation for 1 h, the supernatant was removed, and beads were washed two times with 2 M urea/50 mM TRIS-HCl pH = 7.5, and then the supernatant was combined with the washes. Peptides were reduced with 4 mM DTT for 30 min at 25°C and alkylated by 10 mM iodoacetamide (Sigma-Aldrich) for 45 min in the dark in a thermo shaker. To complete digestion, 0.5 µg trypsin was added to each sample and digested overnight at 25°C. Following this, peptides were desalted on Pierce Peptide Desalting Spin Columns (Thermo Fisher Scientific) and concentrated in a SpeedVac concentrator.

NanoHPLC-HRMS analysis was performed using a nanoRSLC system (ultimate 3000; Thermo Fisher Scientific) coupled to an Easy Exploris 480 (Thermo Fisher Scientific). Peptide separation was carried out using the Easy-nLC ultrahigh-performance LC system. 5 µl of peptide solution was injected and concentrated on a µ-Precolumn Cartridge Acclaim PepMap 100 C18 (i.d. 5 mm, 5 mm, 100 A˚; Thermo Fisher Scientific) at a flow rate of 10 ml/min and using solvent containing H$_2$O/ACN/TFA 98%/2%/0.1%. Peptide separation was performed on a 75-mm i.d. × 500 mm (2 µm, 100 A˚) PepMap RSLC C18 column (Thermo Fisher Scientific) at a flow rate of 300 nl/min. Solvent systems were as follows: (A) 100% water, 0.1%FA, (B) 100% acetonitrile, 0.1% FA. The following gradient was used $t$ = 0 min 2% B; $t$ = 3 min 2%B; $t$ = 103 min, 20% B; $t$ = 123 min, 32% B; $t$ = 125 min 90% B; $t$ = 130 min 90% B; (temperature was regulated at 40°C). MS spectra were acquired at a resolution of 120,000 (200 m/z) in a mass range of 375–1500 m/z with an AGC target 3e6 value of and a maximum injection time of 25 ms. The 20 most intense precursor ions were selected and isolated with a window of 2 m/z and fragmented by HCD (Higher energy C-Trap Dissociation) with a normalized collision energy (NCE) of 30%. MS/MS spectra were acquired in the ion trap with an AGC target 5e5 value, the resolution was set at 15,000 at 200 m/z combined with an injection time of 22 ms.

## Mass spectrometry data analyses

MS data were subjected to LFQ analysis using MaxQuant v1.6.17.0 (https://www.maxquant.org/) using the MaxQuant platform (v1.6.7.0). A database search of the MS/MS data was performed in MaxQuant using the Andromeda search engine against *Canis lupus familiaris* UniProtKB database (Nov. 2020)

and MaxQuant contaminants database. Digestion mode was set to Trypsin/P specificity, with a fixed carbamidomethyl modification of cysteine, and variable modifications of protein N-terminal acetylation and methionine oxidation. Mass deviation was set to 20 ppm for the first and 6 ppm for the main search and the maximum number of missed cleavages was set to 2. Peptide and site false discovery rates (FDR) were set to 0.01, respectively. The search for cofragmented peptides in the MS/MS spectra was enabled ("second peptides" option). Protein identification was performed using a minimum of two unique and razor peptides. Quantification was achieved using the LFQ (Label-Free Quantification) algorithm. Razor and unique peptides were used for LFQ quantification and the "label min. ratio count" was fixed at 2. The match between runs option was enabled, allowing a time window of 0.7 min to search for already identified peptides in all obtained chromatograms.

For surface proteomics, following the imputation of missing values by LFQ analyst (Shah et al., 2020), LFQ intensity values were used to calculate Log$_2$Fold change and –Log$_{10}$P values using Welch's $t$ test. For proximity ligation, raw data was analyzed by LFQ analyst using 0.05 as a P value cutoff, Benjamini Hochberg-type FDR correction and Perseus-type missing value imputation.

## Immunoblotting

For immunoblotting, whole cell lysates were obtained by lysing the cells in RIPA buffer containing cOmplete Protease Inhibitor Cocktail (Roche). Following sonication, samples were pelleted, and supernatant was used as whole cell extracts. The protein concentration of each sample was determined as described before and a total amount of 20 µg protein/sample was resolved on 4–20% gradient gels (Bio-Rad). Proteins were transferred onto nitrocellulose or PVDF membranes further blocked with 5% non-fat dry milk-TBST. To reveal biotinylated proteins, the membranes were probed with HRP-conjugated Streptavidin-HRP diluted in 3% BSA-TBST. For antibody reactions, samples were transferred to PVDF. Following blocking, primary antibodies were diluted in 1% non-fat dry milk-TBST and incubated with the membranes overnight at 4°C. HRP-conjugated secondary antibodies were applied and then the signal was developed by a chemiluminescent HRP substrate (Millipore) and detected with a Fusion FX7 instrument (Vilber Loumat). Antibodies and their respective working dilutions used for immunoblotting are summarized in Table S3.

## Bioinformatics, data analyses, and statistics

For correlation analyses, we extracted EMT marker as well as ORP expression data from publicly available TCGA data using https://www.cbioportal.org (Cerami et al., 2012). For ORP expression analyses, a similar approach was used which was described elsewhere (Rajapakse et al., 2018). For cell line expression data, we used The Cancer Cell Line Encyclopedia set (Barretina et al., 2012), and a TCGA dataset was used to analyze lung adenocarcinoma patient data (Hoadley et al., 2018). A previously identified EMT signature gene set was used to define EMT characteristics (Kohn et al., 2014). To calculate the epithelial–mesenchymal index the following method was applied: first, mean expression values for epithelial and

mesenchymal markers were obtained (Mean$_{mes}$ and Mean$_{epi}$), then a cell line/sample-specific EMT value was calculated by subtracting Mean$_{epi}$ from Mean$_{mes}$ and multiplying the value by −1. This formula was used to calculate a value for all cell lines/tumor samples, then each value was normalized to the mean EMT value of the total data set. Data visualization and correlation analyses were carried out using ggplot2 with GGally extension and heatmap.plus package in Rstudio.

To obtain publicly available survival data we used the online tool https://www.kmplot.com (Győrffy, 2021). Briefly, expression data for OSBP and OSBPL2 was analyzed in 719 patient samples identified histologically as lung adenocarcinoma. Patients were split by median expression to obtain high and low expression categories. For data visualization and data analyses, we used ggsurvplot function in ggplot2 using Rstudio. To assess histology data we used the human proteome atlas (Uhlen et al., 2015).

K-means clustering of significantly changed proteins in the surface proteomics data was performed in Rstudio following the guideline reported previously (Rahmatbakhsh et al., 2021) using appropriate packages. Enrichment analysis was performed in Rstudio using clusterProfiler by Bioconductor or gprofiler2 packages (Wu et al., 2021; Raudvere et al., 2019). To identify ER, Golgi, and plasma membrane resident proteins, functional annotation was performed using Panther Classification System and SubcellulaRVis (https://www.pantherdb.org; Watson et al., 2022; Mi et al., 2019).

Raw data analyses and plotting were carried out in Rstudio using ggplot2 and other appropriate packages. Superplots were used to show data obtained from repeated experiments or cysts and were prepared according to the guidelines published elsewhere (Lord et al., 2020). On these plots, each color represents data obtained from one experiment/one cyst and filled dots/triangles correspond to mean values. Mean values between conditions were compared by appropriate statistical tests indicated in the figure legends. Differences were considered statistically significant when P values were <0.05.

Experiments shown in Fig. 1, F, G, L, and M; and Fig. S1, D and E were repeated three times with similar results. Live imaging experiments are shown in Fig. 1, J and K; and Fig. 2, A–D; and Fig. 6, A and B were repeated three times with similar results. All repeats of the live imaging experiments are shown side-by-side in the corresponding supplementary figures. Experiments with NR12A (Fig. 6, D–F) were repeated three times. Each repeat and statistics obtained for the three independent experiments are summarized in Fig. S5.

### Online supplemental material

Fig. S1 shows the effects of PI4KIIIbeta inhibition on TGN-recruitment of OSBP, apicobasal cargo sorting, and cell polarity establishment. It also shows the three independent RUSH experiments presented in Fig. 1 and presents more representative cysts showing morphology defects upon PIK94 treatment and OSBP silencing. Fig. S2 shows all three independent experiments performed by various RUSH cargo molecules and it also shows more representative cysts upon SWG treatment. Fig. S3 presents the design and the additional experiments performed for the proteomics experiments shown. Fig. S4 presents

more representative D4H-EGFP-expressing cysts upon SWG treatment and OSBP silencing. It also presents the three independent experiments performed with the NR12A dye and their statistical summary. Fig. S5 contains the three independent experiments performed with the P4M-SidM-expressing MDCK and A549 cells. It also shows additional bioinformatic analysis performed on clinical samples. Video 1 shows typical MDCK morphology changes upon DMSO or SWG administration. Table S1 contains data used to analyze the proximity biotinylation experiment. Table S2 contains data used to analyze the surface proteomics experiment. Table S3 contains detailed information about the materials used in the study.

### Data availability

The mass spectrometry proteomics data have been deposited to the ProteomeXchange Consortium via the PRIDE (Perez-Riverol et al., 2022) partner repository with the dataset identifier PXD046322 and https://doi.org/10.6019/PXD046322. All scripts used for data analysis and visualization will be provided upon request.

## Acknowledgments

We are grateful to Frédéric Brau, Julie Cazareth, and Nathalie Leroudier for their technical contribution. We thank Fanny Roussi for providing SWG.

This work was supported by la Fondation pour la Recherche Médicale (DEQ20180339156) and the European Research Council (856404—SPHERES). We are grateful to Stéphanie Miserey for providing the GP135-RUSH model cargo-expressing construct. A postdoctoral fellowship to D. Kovács was provided by Agence National de la Recherche within the project entitled Investissements d'Avenir UCA$^{JEDI}$ (ANR-15-IDEX-01).

Author contributions: D. Kovács, B. Antony, B. Mesmin, and F. Luton designed and conceptualized the research project. B. Antony supervised the work. D. Kovács carried out the experiments. A. Patel contributed to cloning design and molecular biology. A.-S. Gay and D. Debayle performed mass spectrometry and proteomics. D. Kovács and B. Antony analyzed the data and wrote the manuscript.

Disclosures: The authors declare no competing interests exist.

Submitted: 11 July 2023

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

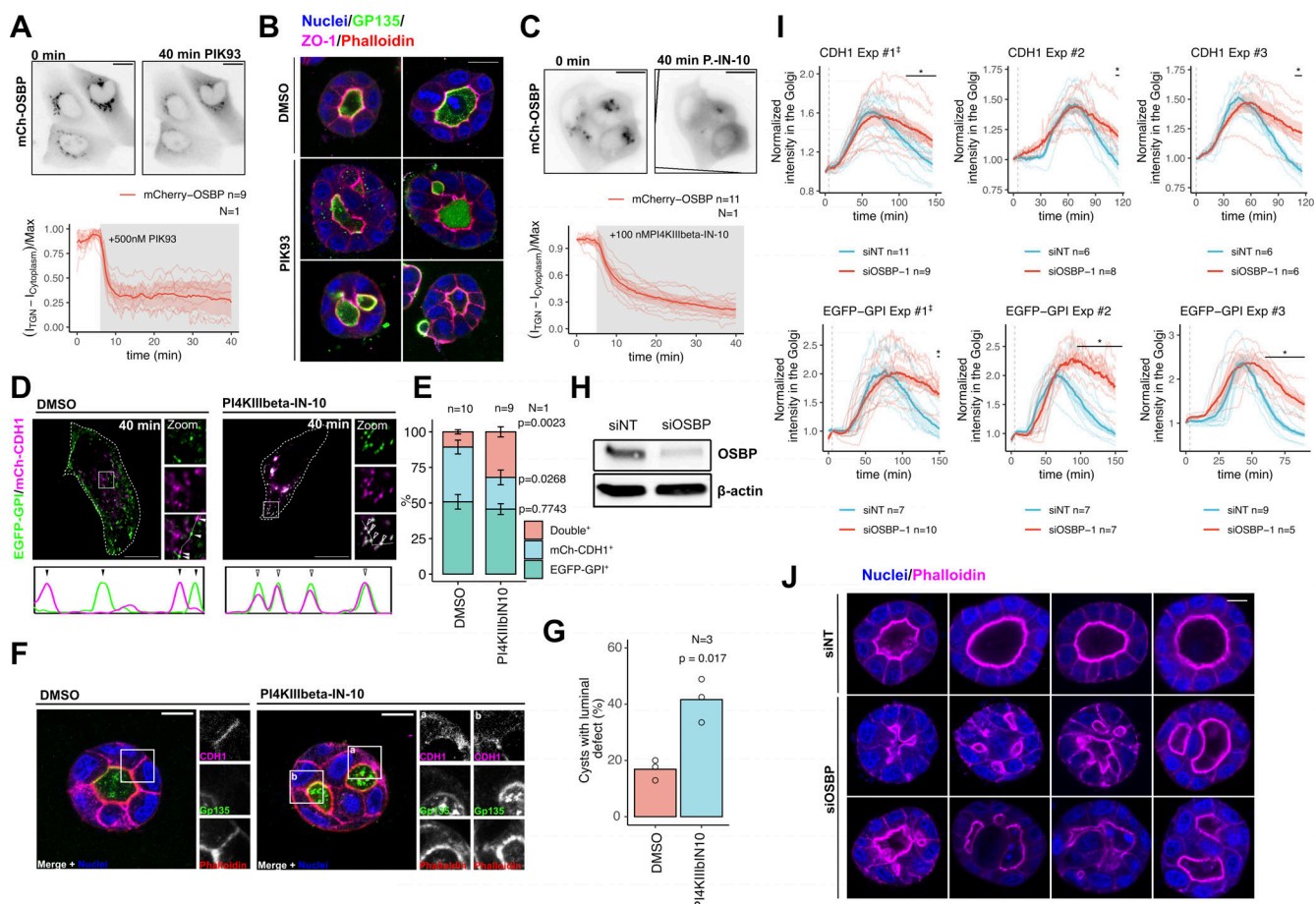

Figure S1. **Effects of PI4KIIIβ inhibition and OSBP silencing on cargo sorting, trafficking, and cyst morphology. (A)** MDCK cells were transfected with mCherry-OSBP-expressing plasmid and mounted for epifluorescence time-lapse microscopy the next day. PIK93 was added to the cells for 5 min at 500 nM concentration. PIK93 treatment reduced the TGN-associated OSBP pool by 75%. N = number of independent experiments presented, n = number of cells analyzed. Bar = 10 μm. **(B)** Additional images showing the effect of PIK93 treatment on MDCK cyst morphology. **(C)** MDCK cells were nucleofected with mCherry-OSBP-expressing plasmid and time-lapse fluorescence imaging was performed on the next day. PI4KIIIbeta-IN-10 was added to the cells after 5 min at 100 nM concentration. PI4KIIIbeta-IN-10 reduced the TGN-associated Golgi pool by ~70%. N = number of independent experiments presented, n = number of cells analyzed. Bar = 10 μm. **(D and E)** MDCK cells were transfected with basolateral CDH1 and apical EGFP-GPI RUSH model cargo-expressing constructs. 24 h later, biotin was added to the samples, and then cells were kept at 19.5°C for 1.5 h to synchronize cargoes at the TGN. Following this, cells were kept at 37°C for 40 min in the absence or presence of 100 nM PI4KIIIbeta-IN-10. After the treatment, cells were fixed, then confocal microscopy was performed. Line-plots indicate fluorescence intensities of each cargo at the corresponding areas and arrowheads point to post-Golgi vesicles positive for single or double cargoes. **(E)** Distribution of each single- and double cargo-containing post-Golgi vesicles in control and PI4KIIIbeta-IN-10-treated cells. At least 9 cells/condition were quantified; in each cell, at least 30 post-Golgi vesicles were counted. N = number of independent experiments presented, n = number of cells/sample used for the analysis. Error bars show SEM and P values were calculated by two-way ANOVA followed by Sidak's multiple comparison test. Bar = 10 μm, diameter of the insets = 9 μm. **(F and G)** MDCK cells were seeded into Matrigel matrix and cysts were left to develop for 72 h in the absence or the presence of 25 nM PI4KIIIbeta-IN-10. **(I)** The number of cysts with luminal defects increased upon PI4KIIIbeta-IN-10 treatment. N = number of independent experiments. P value was calculated by unpaired t test. Bar = 10 μm, diameter of the insets = 14.5 μm. **(H)** Western blot analysis to test the efficacy of OSBP silencing following transfection with RUSH cargoes. **(I)** Results of three independent RUSH experiments using CDH1 and EGFP-GPI model cargoes upon OSBP silencing. Thin lines indicate individual cells. Bold lines correspond to means, while shaded areas indicate SEM. n = number of cells analyzed/experiment to obtain mean and SEM. P values were calculated by two-way ANOVA followed by Sidak's multiple comparison test. ‡ indicates the representative experiments shown in Fig. 1, J and K, respectively. **(J)** Additional cysts to show the effect of OSBP silencing on MDCK cyst development. Bar = 10 μm. Source data are available for this figure: SourceData FS1.

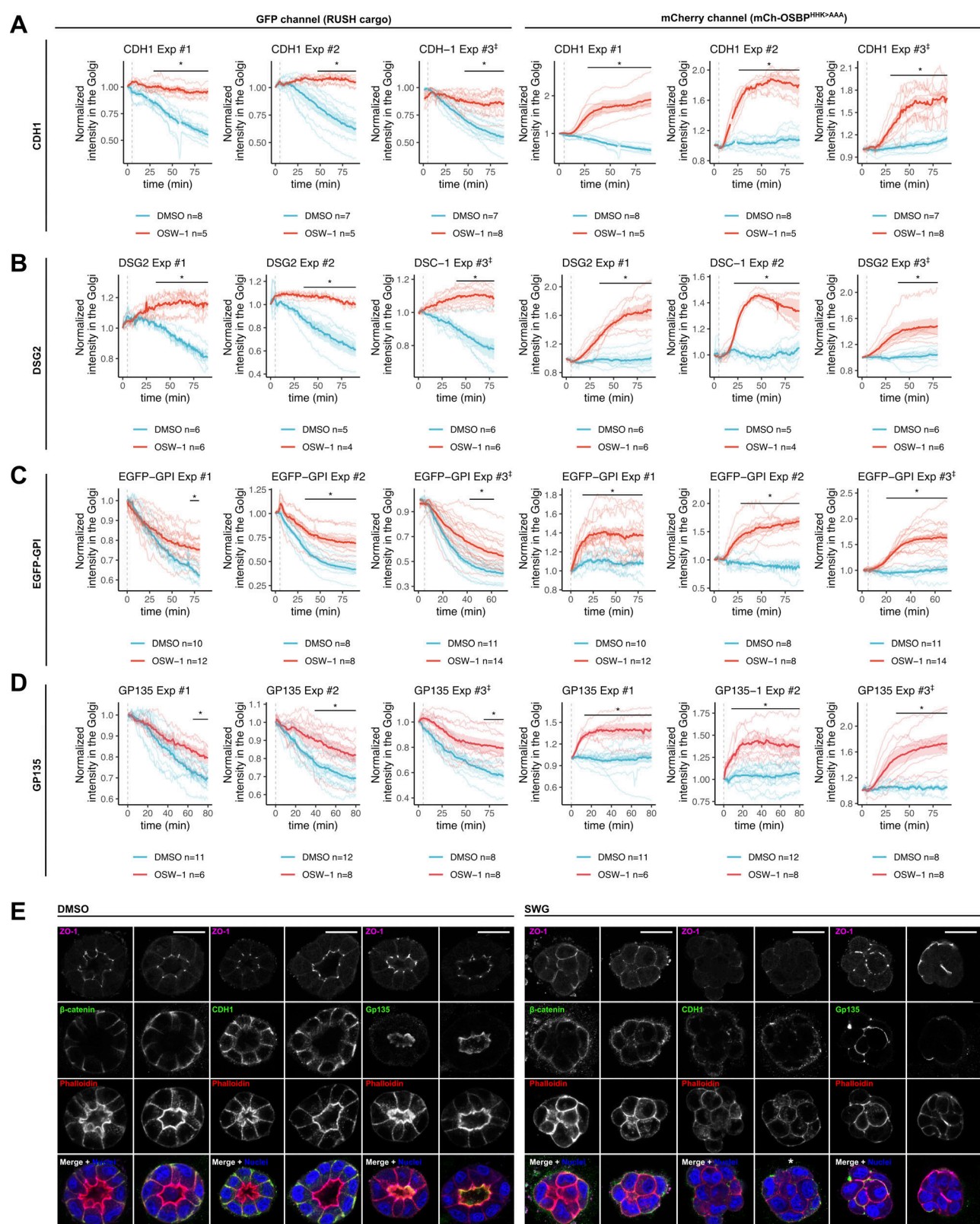

Figure S2. **Effects of OSBP inhibition on apical and basolateral cargo trafficking and cyst morphology. (A–D)** Results of three independent experiments using CDH1, DSG2, EGFP-GPI, and GP135 RUSH model cargoes co-transfected with mCherry-OSBP^HHK>AAA upon OSW-1 treatment. Similar to Fig. 2, A–D, the fluorescence intensities obtained for the two detection channels (EGFP-cargoes, mCherry-OSBP^HHK>AAA) are shown on separate plots. Thin lines indicate individual cells. Bold lines correspond to means, while shaded areas indicate SEM. *n* = number of cells analyzed/experiment to obtain mean and SEM. *P < 0.05. P values were calculated by two-way ANOVA followed by Sidak's multiple comparison test. ‡ indicates the representative experiments shown in Fig. 2, A–D. **(E)** Additional images to show the effect of SWG treatment on MDCK cyst morphology. * indicates cyst shown in Fig. 2 E. Bar = 20 µm.

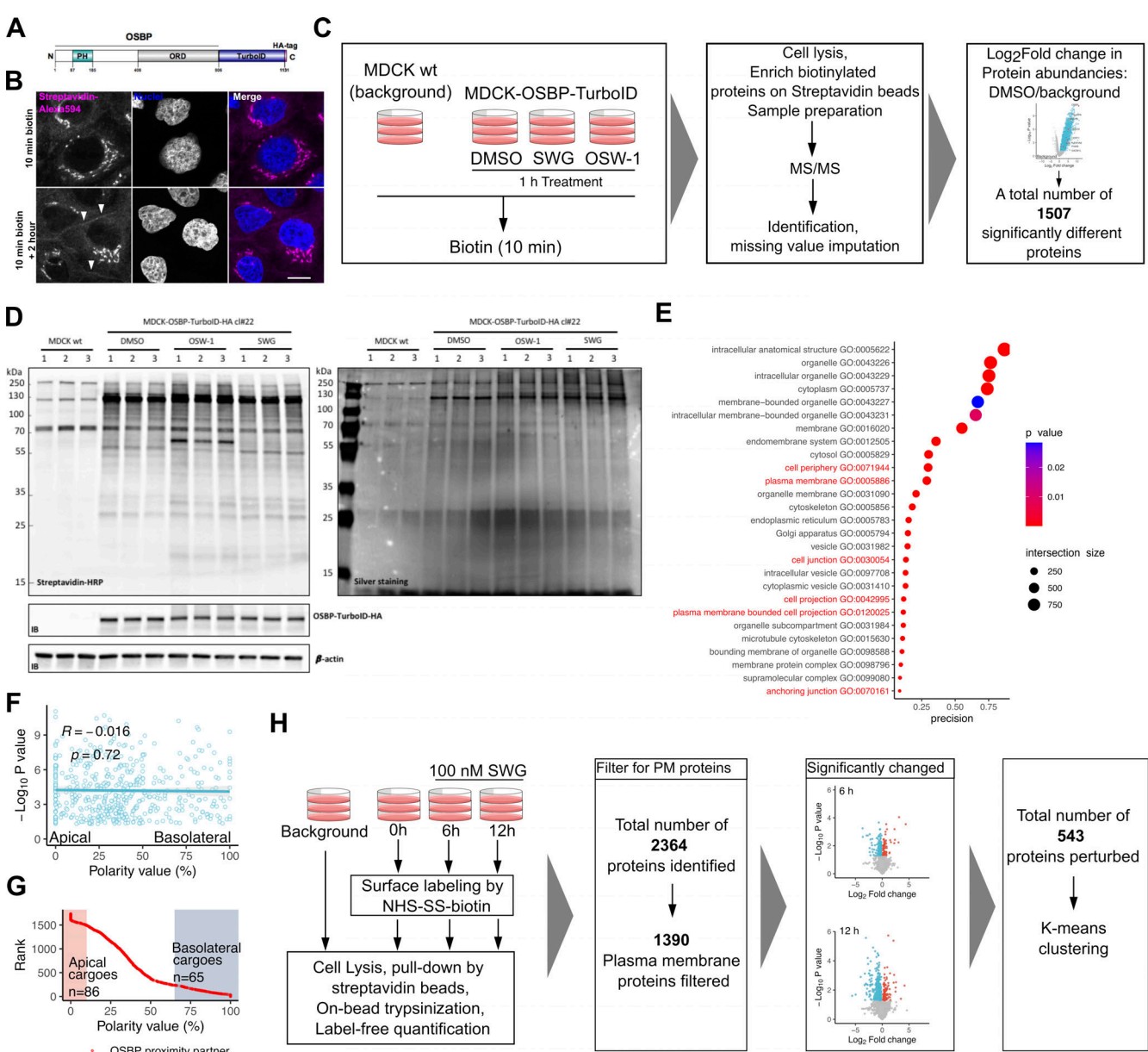

Figure S3. **OSBP proximity labelling and cell surface proteomics. (A)** Domain organization of the OSBP-TurboID-HA construct stably expressed in MDCK cells. **(B)** Confocal imaging shows that incubating OSBP-TurboID expressing MDCK cells for 2 h after biotin treatment results in the enrichment of biotinylation signal at cell-cell junctions (arrowheads). Bar = 12 μm. **(C)** Experimental design and data analysis strategy of the TurboID experiment. **(D)** Biotinylation profiles of cell lysates obtained from wild-type MDCK or MDCK-OSBP-TurboID-HA cells. Membranes probed by streptavidin-HRP confirm successful protein bio-tinylation in MDCK cells stably expressing OSBP-TurboID-HA. Note that ORPphilin treatments affect the biotinylation profile (left panel). Elution profiles of streptavidin beads incubated with lysates obtained from wild-type MDCK or MDCK-OSBP-TurboID-HA cells. Biotinylated proteins were enriched on strep-tavidin beads; then beads were eluted by boiling, elution fractions were resolved on SDS-PAGE and revealed by silver staining (right panel). Elution profiles are comparable to biotinylation profiles shown in the left panel. **(E)** GO cellular component enrichment analyses show significant enrichment in multiple GO classes. **(F)** No correlation between polarity value and abundancy in the OSBP proximity proteome can be established, indicating that cargoes with both apical and basolateral distribution can be identified among the OSBP proximity partners. **(G)** Surface proteins identified by Caceres et al. (2019) were ranked ac-cording to their distribution on the polarized MDCK cell surface (Polarity value). Red dots represent surface proteins identified also in the OSBP proximity proteome. Based on this ranking, highly apical and highly basolateral proteins were selected and subsequently mapped to the OSBP proximity proteome as shown in Fig. 3, G and H. **(H)** Experimental design and data analysis strategy applied for the surface proteomics experiment presented in Fig. 4. Source data are available for this figure: SourceData FS3.

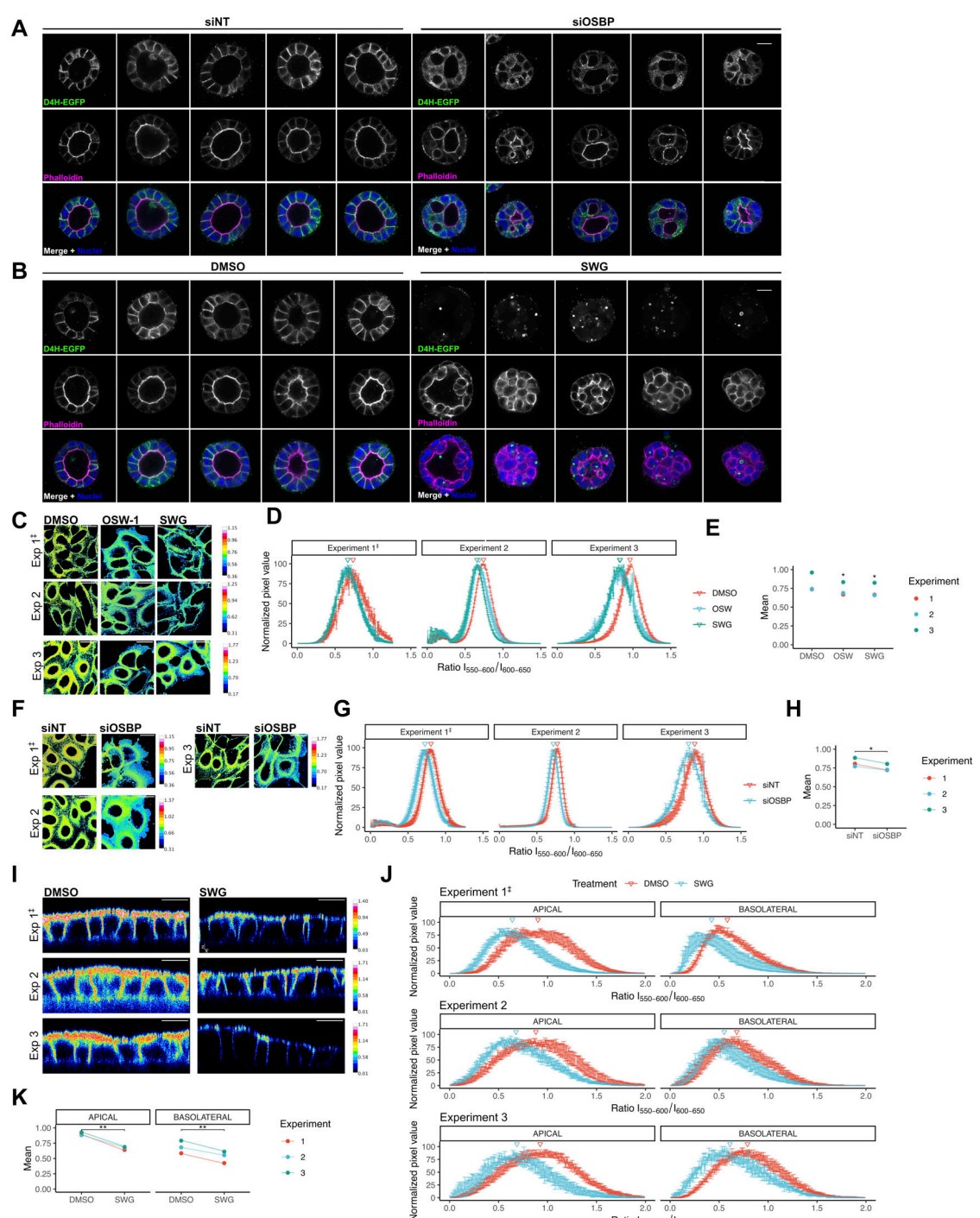

Figure S4. **D4H-EGFP distribution and lipid order imaging in MDCK cells upon OSBP silencing and inhibition. (A and B)** Additional cysts to show the effect of OSBP silencing (A) or SWG-treatment (B) on D4H-GFP distribution. Bar = 10 µm. **(C–E)** Results of three independent experiments on the effect of SWG-treatment on MDCK plasma membrane lipid order. **(C)** Representative pseudo-colored micrographs of each independent experiment. **(D)** Histograms showing ratio values in DMSO and SWG-treated MDCK cells across the three repeats. **(E)** The mean values obtained from each independent experiment were compared by one-way ANOVA with Dunnett's post hoc test. *P < 0.05. Histograms show the mean ± SEM of four fields, each containing at least 8–10 cells. Triangles point to mean values. Bar = 25 µm. ‡ indicates the representative experiment shown in Fig. 5 D. **(F–H)** Results of three independent experiments on the effect of OBP silencing on MDCK plasma membrane lipid order. **(F)** Representative pseudo-colored micrographs of each independent experiment. **(G)** Histograms showing ratio values in control and OSBP-silenced MDCK cells across the three repeats. **(H)** The mean values obtained from each independent experiment were compared by paired $t$ test. *P < 0.05. Histograms show the mean ± SEM of four fields, each containing at least 8–10 cells. Triangles point to mean values. Bar = 25 µm. ‡ indicates the representative experiment shown in Fig. 5 E. **(I–K)** Results of three independent experiments on the effect SWG treatment on polarized MDCK plasma membrane lipid order. **(I)** Representative pseudo-colored micrographs of each independent experiment. **(J)** Histograms showing domain-specific ratio values in control and SWG-treated MDCK cells across the three repeats. **(E)** The mean values obtained from each independent experiment were compared by paired $t$ test. *P < 0.05, **P < 0.01. Histograms show the mean ± SEM of four fields, each containing at least 8–10 cells. Triangles point to mean values. Bar = 10 µm. ‡ indicates the representative experiment shown in Fig. 5 F.

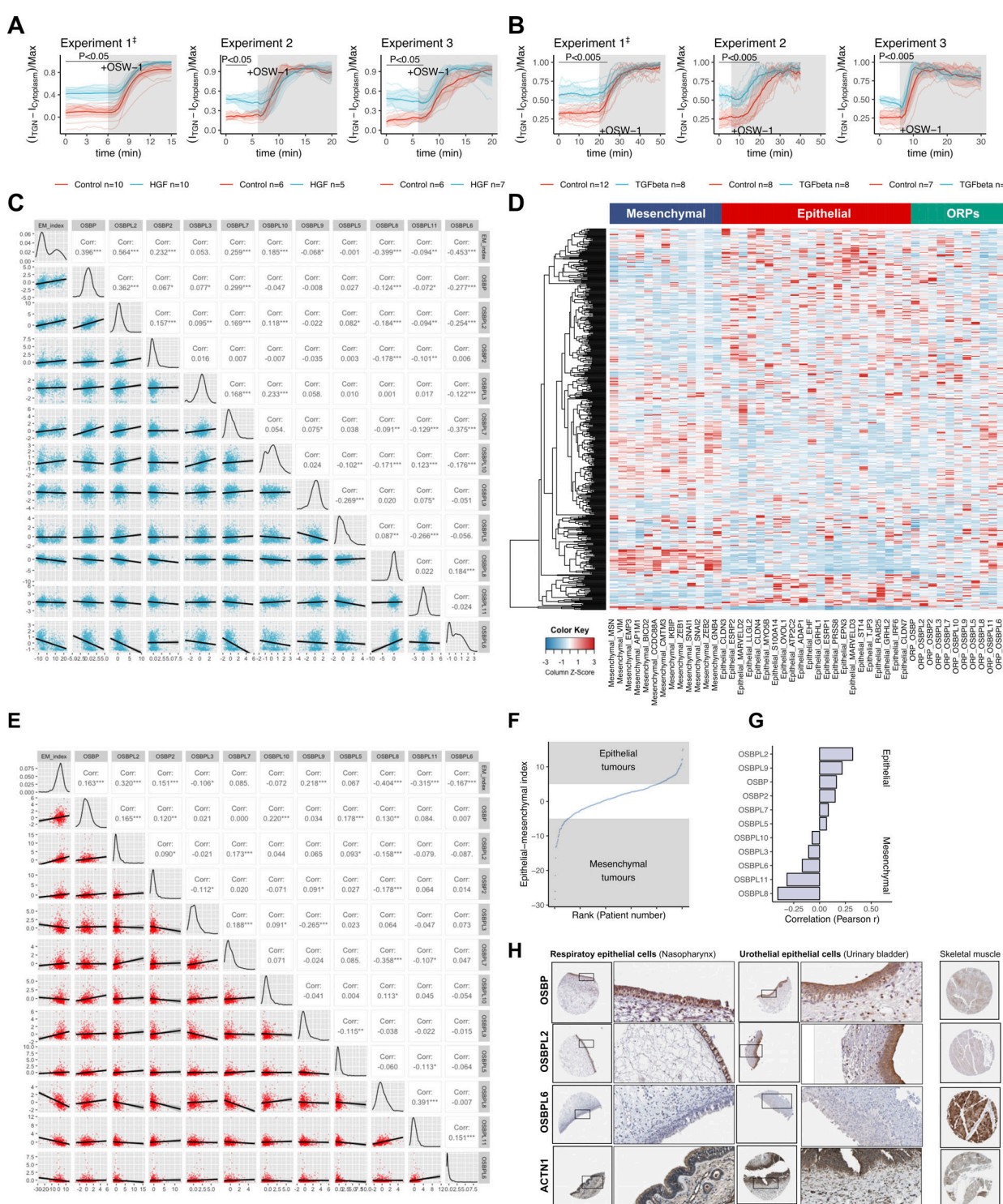

Figure S5. **OSBP-dependent PI(4)P consumption in epithelial and mesenchymal cells and meta-analysis of OSBP-expression. (A and B)** Results of three independent live imaging experiments using MDCK-PI4M-SidM (A) and P4M-SidM-expressing A549 cells (B). Thin lines indicate individual cells. Bold lines correspond to means, while shaded areas indicate 95% confidence interval. *n* = number of cells analyzed/experiment to obtain mean and 95% CI. P values were calculated by two-way ANOVA followed by Sidak's multiple comparison test. ‡ indicates the representative experiments shown in Fig. 5, A and B, respectively. **(C)** Correlation matrix comparing ORP expression values and epithelial-mesenchymal index corresponding to cell lines included in the Cancer Cell Line Encyclopedia. Corr = Pearson r, *P < 0.05, **P < 0.01, ***P < 0.005. **(D)** Gene expression heatmap containing expression values for mesenchymal, epithelial marker genes as well as ORP genes of TCGA lung adenocarcinoma cohort. **(E)** Correlation matrix comparing ORP expression values and epithelial–mesenchymal indexes corresponding to tumor specimens included in the TCGA lung adenocarcinoma cohort. Corr = Pearson r, *P < 0.05, **P < 0.01, ***P < 0.005. **(F)** Calculated epithelial–mesenchymal indexes corresponding to each lung adenocarcinoma patient sample. **(G)** Correlations between epithelial-mesenchymal index and ORP expression values in TCGA lung adenocarcinoma samples. **(H)** Histology samples from The Human Protein Atlas show high epithelial-specific OSBP and OSBPL2 staining.

Video 1.   **Time-lapse epifluorescence imaging of stable EGFP-expressing MDCK cells treated with DMSO or 100 nM SWG.** SWG treatment induces the scattering of MDCK colonies. DMSO/SWG was added to the cells at time 0. EGFP intensity was recorded in 1 image/15 min during acquisition. Video display rate: 20 frames/s. Scale bar = 150 µm. Related to Fig. 2 F.

**Provided online are Table S1, Table S2, and Table S3. Table S1 contains data used to analyze the proximity biotinylation experiment. Table S2 contains data used to analyze the surface proteomics experiment. Table S3 contains detailed information about the materials used in the study.**

