## [Peer Review File · The Journal of Cell Biology]

Lipid exchange at ER-trans Golgi contact sites governs polarized cargo sorting

Dávid Kovács, Anne-Sophie Gay, Delphine Debayle, Sophie Abélanet, Amanda Patel, Bruno Mesmin, Frédéric Luton, and Bruno Antonny

Corresponding Author(s): Bruno Antonny, Institut de Pharmacologie Moléculaire et Cellulaire

Review Timeline:

Submission Date:	2023-07-11
Editorial Decision:	2023-08-15
Revision Received:	2023-10-18
Editorial Decision:	2023-10-20
Revision Received:	2023-10-26

Monitoring Editor: William Prinz

Scientific Editor: Andrea Marat

Transaction Report:

DOI: <https://doi.org/10.1083/jcb.202307051>

August 15, 2023

Re: JCB manuscript #202307051

Dr. Bruno Antony
Université Côte d'Azur et CNRS
Institut de Pharmacologie Moléculaire et Cellulaire
660 route des Lucioles
Valbonne 06560
France

Dear Dr. Antony,

Thank you for submitting your manuscript entitled "Lipid exchange at ER-trans Golgi contact sites governs polarized cargo sorting and secretion". The manuscript was assessed by expert reviewers, whose comments are appended to this letter. We invite you to submit a revision if you can address the reviewers' key concerns, as outlined here.

You will see that the reviewers are overall positive about the potential significance of your results and have provided constructive feedback and suggestions. Most of their comments can be addressed via discussion and clarifications to your text. However, we expect you to provide new data to address the specific experimental concerns of reviewer 2 and 3.

GENERAL GUIDELINES:

Text limits: Character count for an Article is < 40,000, not including spaces. Count includes title page, abstract, introduction, results, discussion, and acknowledgments. Count does not include materials and methods, figure legends, references, tables, or supplemental legends.

Figures: Articles may have up to 10 main text figures. Figures must be prepared according to the policies outlined in our Instructions to Authors, under Data Presentation, <https://jcb.rupress.org/site/misc/ifora.xhtml>. All figures in accepted manuscripts will be screened prior to publication.

Supplemental information: There are strict limits on the allowable amount of supplemental data. Articles may have up to 5 supplemental figures. Up to 10 supplemental videos or flash animations are allowed. A summary of all supplemental material should appear at the end of the Materials and methods section.

Please note that JCB now requires authors to submit Source Data used to generate figures containing gels and Western blots with all revised manuscripts. This Source Data consists of fully uncropped and unprocessed images for each gel/blot displayed in the main and supplemental figures. Since your paper includes cropped gel and/or blot images, please be sure to provide one Source Data file for each figure that contains gels and/or blots along with your revised manuscript files. File names for Source Data figures should be alphanumeric without any spaces or special characters (i.e., SourceDataF#, where F# refers to the associated main figure number or SourceDataFS# for those associated with Supplementary figures). The lanes of the gels/blots should be labeled as they are in the associated figure, the place where cropping was applied should be marked (with a box), and molecular weight/size standards should be labeled wherever possible.

The typical timeframe for revisions is three to four months. While most universities and institutes have reopened labs and allowed researchers to begin working at nearly pre-pandemic levels, we at JCB realize that the lingering effects of the COVID-19 pandemic may still be impacting some aspects of your work, including the acquisition of equipment and reagents. Therefore, if you anticipate any difficulties in meeting this aforementioned revision time limit, please contact us and we can work with you to

find an appropriate time frame for resubmission. Please note that papers are generally considered through only one revision cycle, so any revised manuscript will likely be either accepted or rejected.

Thank you for this interesting contribution to Journal of Cell Biology. You can contact us at the journal office with any questions, cellbio@rockefeller.edu or call (212) 327-8588.

Sincerely,

William Prinz, PhD
Monitoring Editor

Andrea L. Marat, PhD
Senior Scientific Editor

Journal of Cell Biology

Reviewer #1 (Comments to the Authors (Required)):

Kovács et al investigated the influences of PI4 kinase signaling and OSBP function in the sorting of secretory protein cargo upon exit from the Golgi complex. The studies were mostly conducted with cultured RPE-1 cells and mesenchymal and polarized epithelial MDCK cells. The authors report that oscillating waves of a PI4P reporter on Golgi membranes are correlated with export of an ER-anchored RUSH GFP-GPI reporter protein from the Golgi. After exit from the Golgi, GFP-GPI, an apical targeted protein, and CDH1, a basolateral resident protein, are present in different populations of Golgi-derived cytoplasmic vesicles. Inhibition of PI4K by PIK93 resulted in accumulation of double positive (GFP-GPI plus CDH1) vesicles, suggesting that PI4K is required for the sorting of apical from basolateral proteins in these cells. Similarly, depletion of OSBP in MDCK cyst cultures using RNAi or chemical inhibitors showed effects indicative of disrupted apical-basolateral polarity. This could be attributed to aberrant localization/trafficking of many of apical and basolateral resident proteins upon loss of OSBP activity, though the greatest effect was observed for sorting to the basolateral membrane (eg, CDH1, DSG2). The results suggest that OSBP is a primary effector of PI4K signaling that is relevant to apical-basolateral cargo protein sorting upon exit from the Golgi. Spatial aspects of OSBP-dependent cargo sorting were investigated using proximity biotinylation experiments that were designed to identify proteins, including secreted cargo proteins, that are in close proximity to OSBP during trafficking through the early secretory pathway (the 'OSBP proximity proteome'). The results show that sorting of basolateral proteins is most affected by depletion/inhibition of OSBP. Finally, it is shown that loss of OSBP function results in depletion of cholesterol from the plasma membrane and its accumulation on other organelle membranes.

This is an interesting study that implicates OSBP-dependent trafficking of cholesterol/PI4P in apical-basolateral of plasma membrane proteins in MDCK cells. Given the prominent role that OSBP plays in trafficking cholesterol and PI4P throughout the cell, it's not surprising (at least to me) that perturbation to OSBP function would cause wide-spread effects on lipid and protein sorting, but this study addresses these questions with innovative approaches. And I'm surprised that the major effects are on targeting to the basolateral plasma membrane, given the abundance of cholesterol in the apical membrane.

I find the results of the OSBP-proximity biotinylation (Fig. 3) experiments, which constitute one of the foundations of the study, only modestly compelling. I question if the OSBP proximity proteome described here, is such a meaningful distinction, as elaborated below.

1. Over 1500 proteins were enriched in the OSBP proximity proteome, including several established OSBP-interacting proteins (labelled in the figure), but many, many proteins were enriched in this proteome and many of the expected hits were not convincingly distinguished from other irrelevant proteins (not labelled in the figure).
2. How does the biotinylation assay distinguish between the ER-localized 'neo-pool' of plasma membrane proteins from the Golgi-localized 'trafficking pool'? At the ER-Golgi MCS, 'trafficking' proteins in both the ER and the Golgi compartments are candidate substrates for OSBP-TurboID, provided that TurboID can access ER proteins, which the VAP protein hits demonstrate. Given the C-terminus location of TurboID and also because OSW or SWG were used, this complicates interpretation.
3. ORPphillin treatment depletes the cytoplasmic pool OSBP due to its accumulation in a "crowded, non-dynamic MCSs along the ER-TGN interface", which raises the issues of specificity and relevance of the biotinylation experiments that arising potentially from changes in the size and dynamic properties of the ER-Golgi MCS. Are the OSBP proximity interactomes of untreated cells the same, or different?
4. Regarding the application of the biotinylation results to the sorting of cell surface proteins (ie, cargo proteins), proximity to

OSBP should be influenced significantly by the relative abundances of the secretory protein cargos and this has important implications for interpreting the results of sorting assays. Without knowing the proportions of apical vs basolateral proteins in the cells, sorting fidelity cannot be confidently established in the treated cells. In addition, these experiments would likely have been more informative if polarized cells were used.

5. Figure 4 shows that the intracellular localization of the D4H cholesterol probe is quite different in OSBP-depleted cells vs SGW-treated cells. Shouldn't depletion of OSBP by RNAi and inhibition of OSBP by SWG result in the same distribution? Why doesn't it?

Minor points:

6. Figure 3, panels E-L: I don't understand what colors (red, grey, blue) are used to indicate the three classes of proteins. In the volcano plots (3E and G), which proteins class is shown in red. Please clarify.

7. The manuscript should be checked for misspelled words. "Time lapse" is often spelled "time laps", "fluorescent" is frequently used where "fluorescence" should be used and "wysts" is used instead of "cysts".

Reviewer #2 (Comments to the Authors (Required)):

The paper shows a new role of OSBP in cargo sorting in polarized epithelial cells. OSBP localizes to ER/TGN membrane contact sites (MCS), extracts cholesterol from the ER membrane, and transfers it to TGN membranes. The transfer activity consumes PI4P, enriched in TGN membranes produced by PI4-kinases. Depletion of OSBP has a significant impact on protein sorting in the trans-Golgi Network (TGN). This is manifested by missorting different plasma membrane proteins and losing polarity. The authors suggest that OSBP generates lipid gradients in the TGN required for cargo sorting.

The experiments have been carefully performed, and the research question is highly significant and exciting for JCB readers. Nevertheless, there are a few questions that should be clarified. The clarity of the text can also be greatly improved.

Major concerns:

1. Please clarify: OSBP regulates apical and basolateral sorting in the TGN. Is it surprising that both cargo types are in proximity to OSBP? Are there cargo types that are excluded? Could all cargo proteins move through this domain initially?

2. In the MS analysis with OSW treatment, the authors identify more basolateral proteins near OSBP and conclude that apical and basolateral cargo must be segregated in the TGN. However, the segregation hypothesis would require confirmation by immunostaining (apical versus basolateral versus OSBP and/or PI4P).

3. What is the significance of hits in OSW-treated cells? You generate a huge domain artificially. Since sorting is a highly dynamic process, how can this domain capture cargo molecules precisely? Please explain.

4. Both apical and basolateral membranes contain cholesterol. Why is the sorting of basolateral proteins much more affected?

5. The authors talk about various secretory routes (line 373). Which routes are they referring to?

Minor comments and suggestions:

Please change Golgi to TGN, be consistent with the expression

Please use cargo, not secretory cargo (you look at membrane-associated proteins)

Please do not use the word secretion for cell surface transport (proteins stay associated with the membrane)

Line 163to monitor cargo export dynamics...

Line 165.... inhibit/impair OSBP-mediated lipid exchange....

Line 168... Please reformulate the sentence. It is confusing.

At 20 degrees, cargo colocalizes with mcherry-OSBP in control cells...

The shift to 37 degrees releases cargo from the TGN while OSBP still localizes to the TGN.

In contrast, treatment of cells with OSW significantly impairs TGN export of basolateral cargo at 37 degrees as demonstrated by a persistent co-localization with OSBP over time.

Line 177...apical and basolateral cargo require OSBP-dependent MCS...

Line 194 - 195 OSBP-engaged MCS govern apical and basolateral sorting routes....

Line 251 domain-selective mass spectrometry. Please describe what it is in a short sentence.

Line 273 ...the abundance...

OSW treatment impaired TGN export of CDH1

Line 340... we directly analyzed..

Reviewer #3 (Comments to the Authors (Required)):

This manuscript investigates the relationship between the TGN-localized PI4P/cholesterol exchanger OSBP and the secretory function of the Golgi in polarized epithelial cells. Genetic or pharmacological disruption of OSBP disrupts export of model apical

and basolateral cargoes from the TGN (with a more severe effect on basolateral), and also impairs their segregation into distinct secretory carriers. It also impairs successful formation and maintenance of polarized cysts in an MDCK model of epithelia formation. A proximity proteomic screen confirms stalling of both apical and basolateral cargoes, with a more profound effect on the latter. OSBP inhibition also prevents cholesterol enrichment in both apical and basolateral membranes, and increases membrane disorder. Finally, the epithelial phenotype is shown to turnover more PI4P and be associated with higher expression of cholesterol-trafficking proteins ORP2 and OSBP - both of which correlate with increased survival of patients with lung adenocarcinoma, suggesting reduced epithelial-to-mesenchymal transition.

The manuscript is well written and the data are clearly presented. A combination of genetic and pharmacological approaches are used, and an array of candidate and non-biased screens are employed. The data are consistent and compelling. Overall, the requirements for OSBP in polarized secretion and the establishment and maintenance of a polarized epithelium are clear. Mechanistically, the requirements for OSBP cholesterol/PI4P exchange for polarized secretion is hazier. It is clearly required for PM cholesterol maintenance, but a killer experiment showing cholesterol enrichment at the TGN facilitating sorting is not shown (nor can the reviewer envision a feasible experiment to show this). That being said, the observation of OSBP lipid traffic playing a central role in polarized secretion and the epithelial phenotype is an important finding, and one that has significance for a wide swathe of the community. Some suggestions to improve the manuscript are listed below:

Conceptual suggestions:

The authors' explanation that OSBP inhibition leads to accumulation of basolateral cargo at the TGN because inhibited, PI4P-bound OSBP accumulates at membrane domains rich in PI4P but poor in cholesterol and disordered seems feasible. My idea is: what if PI4Ks (PI4Kbeta and PI4K2alpha) are also inhibited? Would this not cause OSBP enrichment to stop, and at least equalize the rate of apical and basolateral protein secretion (even though it will likely still block sorting)? Indeed, a requirement for cholesterol in membrane protein sorting at the TGN might explain the key requirement for a TGN/ER cholesterol exchanger when many in the cholesterol field poo-poo the idea in favor of direct lysosome/PM and ER/PM exchange.

The use of PIK93 in the experiments in figure 1H and I is problematic, since this compound is known to inhibit class 1B and class III PI3K, which complicates interpretation of the data. A more selective compound such as PI4KIIIbeta-IN-10 should be employed for these experiments.

Comments on technical aspects:

The methods section makes clear that 500 nM PIK93 was used, but this should be detailed in the figure legends for convenience (e.g. figure 1)

The number of independent experiments is not clear in figure 1J-K, 2A-D and should be stated.

Figures 3C, 4 E, F, H: Welch's t-test is used, but there are multiple comparisons. The data should be compared by one-way ANOVA and a suitable post-hoc comparison test

Figure 5D-F: the manuscript states that the histograms show means of 4-fields with 8-10 cells. However, the number of independent experiments analyzed is not clear. It reads like all four fields could have come from the same experiment

Figure S1A: details of mean, error and n are missing from the legend.

Reviewer #1 (Comments to the Authors (Required)):

Kovács et al investigated the influences of PI4 kinase signaling and OSBP function in the sorting of secretory protein cargo upon exit from the Golgi complex. The studies were mostly conducted with cultured RPE-1 cells and mesenchymal and polarized epithelial MDCK cells. The authors report that oscillating waves of a PI4P reporter on Golgi membranes are correlated with export of an ER-anchored RUSH GFP-GPI reporter protein from the Golgi. After exit from the Golgi, GFP-GPI, an apical targeted protein, and CDH1, a basolateral resident protein, are present in different populations of Golgi-derived cytoplasmic vesicles. Inhibition of PI4K by PIK93 resulted in accumulation of double positive (GFP-GPI plus CDH1) vesicles, suggesting that PI4K is required for the sorting of apical from basolateral proteins in these cells. Similarly, depletion of OSBP in MDCK cyst cultures using RNAi or chemical inhibitors showed effects indicative of disrupted apical-basolateral polarity. This could be attributed to aberrant localization/trafficking of many of apical and basolateral resident proteins upon loss of OSBP activity, though the greatest effect was observed for sorting to the basolateral membrane (eg, CDH1, DSG2). The results suggest that OSBP is a primary effector of PI4K signaling that is relevant to apical-basolateral cargo protein sorting upon exit from the Golgi. Spatial aspects of OSBP-dependent cargo sorting were investigated using proximity biotinylation experiments that were designed to identify proteins, including secreted cargo proteins, that are in close proximity to OSBP during trafficking through the early secretory pathway (the 'OSBP proximity proteome'). The results show that sorting of basolateral proteins is most affected by depletion/inhibition of OSBP. Finally, it is shown that loss of OSBP function results in depletion of cholesterol from the plasma membrane and its accumulation on other organelle membranes.

This is an interesting study that implicates OSBP-dependent trafficking of cholesterol/PI4P in apical-basolateral of plasma membrane proteins in MDCK cells. Given the prominent role that OSBP plays in trafficking cholesterol and PI4P throughout the cell, it's not surprising (at least to me) that perturbation to OSBP function would cause wide-spread effects on lipid and protein sorting, but this study addresses these questions with innovative approaches. And I'm surprised that the major effects are on targeting to the basolateral plasma membrane, given the abundance of cholesterol in the apical membrane.

A: We thank the Reviewer for his/her general comments on the manuscript, which can be summarized as follows: it is not surprising that inhibiting OSBP activity affects protein sorting but it is surprising that basolateral cargoes are more affected by this inhibition than apical cargoes. We agree on both points.

The impact of OSBP on cargo traffic was expected given the central role of OSBP in distributing two key lipids of the Golgi apparatus, PI4P and cholesterol. Nevertheless, it was important to test this hypothesis via a comprehensive analysis of cargoes of different membrane topology and final destination, which is provided by the present study.

The fact that basolateral cargoes are more affected than apical ones was at first glance surprising. However, OSBP is not only a cholesterol transfer protein but also displays membrane binding properties that makes it prone to bind to membranes with low packing and to detach from membranes rich in cholesterol. Thus, although OSBP activity makes TGN membranes prone to segregate apical cargoes due to loading TGN membranes with cholesterol, OSBP does not stay on such membranes but recycles on membranes that are PI(4)P-rich and cholesterol-poor. When OSBP is targeted by ORPphilins, cholesterol transfer stops, and OSBP is stabilized on PI(4)P-rich, cholesterol-poor regions, a membrane

environment that does not favor apical cargo accumulation. Instead, ORPphilin-treatment stabilizes contact sites where basolateral cargoes (and most probably cargoes with no polarized trafficking) are preferentially captured as revealed by a Turbo ID approach. To clarify our points, we re-formulated the corresponding text in the manuscript and added new paragraphs into the discussion.

I find the results of the OSBP-proximity biotinylation (Fig. 3) experiments, which constitute one of the foundations of the study, only modestly compelling. I question if the OSBP proximity proteome described here, is such a meaningful distinction, as elaborated below.

1. Over 1500 proteins were enriched in the OSBP proximity proteome, including several established OSBP-interacting proteins (labelled in the figure), but many, many proteins were enriched in this proteome and many of the expected hits were not convincingly distinguished from other irrelevant proteins (not labelled in the figure).

A: Getting 1500 proteins in a TurboID assay (Fig 3D) is in the range of what has been observed in the original method paper describing TurboID and its application to subcellular regions (see Branon, T. C. et al. Efficient proximity labeling in living cells and organisms with TurboID. *Nat. Biotechnol.* 36, 880–887 (2018)). With mitochondria, ER, or nuclear TurboID constructs, the authors obtained 314, 1180 and 1455 protein hits, respectively. Additionally, we have to keep in mind that OSBP is present not only at the ER/TGN interface but also in the cytoplasm, which necessarily increases the number of possible proximity hits.

Notwithstanding, we agree that the high number of proteins is overwhelming and it would be impossible to label all proteins in a volcano plot. However, we did not assume that the full proteome was meaningful. Instead, what is informative is (1) to compare this proteome with a published apical vs basolateral proteome (Fig S3F) and (2) to see how the proteome changes when we shift OSBP localization and block its activity using ORPphilin (Fig 3E to 3H). Here, the coupling between gene ontology and mass spec data gives some interesting trends (e.g. exclusion of many cytosolic proteins, enrichment in basolateral vs apical cargoes Fig3G-H). Please note that we did not compare different proteins under a given condition (whether they are important or irrelevant hits) because each peptide has unique ionization properties and because the yield of biotinylation depends on the number and accessible Lys residues. This is a general issue in proteomic approaches.

2. How does the biotinylation assay distinguish between the ER-localized 'neo-pool' of plasma membrane proteins from the Golgi-localized 'trafficking pool'? At the ER-Golgi MCS, 'trafficking' proteins in both the ER and the Golgi compartments are candidate substrates for OSBP-TurboID, provided that TurboID can access ER proteins, which the VAP protein hits demonstrate. Given the C-terminus location of TurboID and also because OSW or SWG were used, this complicates interpretation.

A: We chose to append the TurboID probe to the C-terminus of OSBP for two reasons. First the N-terminus of OSBP is intrinsically disordered and has some shield-like properties (Jamecna et al. 2019), which might interfere with the scanning of proteins in the vicinity of OSBP. Second, we considered that the C-terminal ORD was a better choice because it interacts with both the ER and TGN and as such can give a broad view of the neighbors of OSBP when the protein is sandwiched between the two organelles. Although the drawback of this choice is that we could not distinguish ER-localized neo-pool and trafficking proteins, we used other

assays to analyze how OSBP controls cargo trafficking (e.g. immunofluorescence of endogenous proteins (Fig4D-H), RUSH assays and plasma membrane proteome biotinylation – Fig2A-D, Fig4A-C). We now better state in the text that the very fact that the OSBP-TurboID fusion localizes at the interface between the ER and the TGN (like wild-type OSBP) makes it prone to biotinylate both ER and TGN proteins.

3. ORPphilin treatment depletes the cytoplasmic pool OSBP due to its accumulation in a "crowded, non-dynamic MCSs along the ER-TGN interface", which raises the issues of specificity and relevance of the biotinylation experiments that arising potentially from changes in the size and dynamic properties of the ER-Golgi MCS. Are the OSBP proximity interactomes of untreated cells the same, or different?

A: They show many significant differences as indicated by the volcano plots on Fig3E and F. On such plots, \log_2 -transformed fold changes values (ORPphilin/DMSO) are shown on the x axis and the P-value calculated for each protein is represented on the y axis. Many proteins reach high fold change values with significant differences ($P > 0.05$), meaning that they are more or less abundant in the proximity proteome following ORPphilin treatments.

We agree that the interpretation of the biotinylated proteins that are close to OSBP-TurboID upon Orphilin treatments is not trivial as these drugs stabilize contact sites that are no longer dynamic in lipid exchange. We have now underlined this point in the revised text. However, the proteome obtained under such conditions is very informative. First, because it serves as a control for the proteome obtained without ORPphilin, i.e. when OSBP is dynamic and exchanges lipids. For instance, the abundance of cytosolic proteins is reduced upon ORPphilin treatment, reflecting the re-distribution of OSBP (Fig3E, F), whereas ER and Golgi proteins become more abundant. Second, it suggests that prior to lipid exchange (that is, enrichment of the TGN in cholesterol at the expense of PI(4)P), OSBP is closer to basolateral cargoes, which is an important point considering the way OSBP binds to the TGN.

4. Regarding the application of the biotinylation results to the sorting of cell surface proteins (ie, cargo proteins), proximity to OSBP should be influenced significantly by the relative abundances of the secretory protein cargos and this has important implications for interpreting the results of sorting assays. Without knowing the proportions of apical vs basolateral proteins in the cells, sorting fidelity cannot be confidently established in the treated cells. In addition, these experiments would likely have been more informative if polarized cells were used.

A: We agree but we do not see how we can address this point (which is quite general in proteomic studies) as it would require having a quantitative view of all cargoes in term of local quantities at the TGN. This seems an extremely difficult aim, especially in fully polarized cells. That said, we reiterate the fact that the proximity analysis of OSBP is only one approach that we have been using. The other ones allow us to go deeper into the mechanisms.

5. Figure 4 shows that the intracellular localization of the D4H cholesterol probe is quite different in OSBP-depleted cells vs SGW-treated cells. Shouldn't depletion of OSBP by RNAi and inhibition of OSBP by SWG result in the same distribution? Why doesn't it?

A: We agree that the two approaches cause a different redistribution of the D4H probe. Still, they both lead to a decrease in cholesterol at the basolateral membrane. However, the way the two experiments are done is different. When we use siRNA, we first nucleofect MDCK cells to silence OSBP. The next day we re-seed the cells to Matrigel to develop cysts from single

cells, which typically takes 3-4 days. This way, we see the effect of OSBP silencing on the establishment of 3D cysts and on the distribution of cholesterol. However, we cannot rule out that during this time window, other ORP can compensate for the loss of OSBP and partially restore cholesterol transfer.

For drug treatments, D4H expressing MDCK cells are first let to develop cysts, then OSBP is targeted by ORPphilin treatments to block its activity overnight. Thus, here we test the effect of OSBP-targeting on the maintenance of apico-basal polarity. As a consequence of ORPphilin treatments, Golgi exit of cargoes (first basolateral, then apical) is inhibited due to the drastic morphological and functional effects in the Golgi, which eventually culminates in the loss of cell polarity.

Upon short ORPphilin treatments and siRNA-mediated OSBP silencing, we typically find D4H-probe enriched in small lysosomes (Péresse et al. 2020). However, upon extended ORPphilin treatments the probe appears in large intracellular aggregates as shown in the Figure. When cholesterol levels rise in the ER, it is stored as cholesterol ester in ER-derived lipid droplets to avoid cytotoxicity. However, cholesteryl-esters are not recognized by the D4H probe. We think that due to the long ORPphilin treatment, the cholesterol level in cellular membranes drops below the sensitivity threshold of the D4H probe (20 mol%). Given that the probe is constitutively expressed in the cells, it might form large intracellular aggregates.

Minor points:

6. Figure 3, panels E-L: I don't understand what colors (red, grey, blue) are used to indicate the three classes of proteins. In the volcano plots (3E and G), which proteins class is shown in red. Please clarify.

A: We agree that the applied color code was misleading as we used the same color (red) to, highlight either selected proteins in the volcano plots or “upregulated” proteins on the stacked bar plots. Furthermore, apical proteins were labelled blue and basolateral proteins were red in Fig3 I, J, whereas same colors were used to highlight proteins that increase or decrease in the bar plots on J and K. To avoid this misunderstanding, we changed the color codes of the volcano plots on the corresponding figures and now the colors represent the categories (no change, up-, or downregulated) shown on the bar plots.

7. The manuscript should be checked for misspelled words. "Time lapse" is often spelled "time laps", "fluorescent" is frequently used where "fluorescence" should be used and "wysts" is used instead of "cysts".

A: Thank you. We checked the entire manuscript and corrected typos and spelling mistakes.

Reviewer #2 (Comments to the Authors (Required)):

The paper shows a new role of OSBP in cargo sorting in polarized epithelial cells. OSBP localizes to ER/TGN membrane contact sites (MCS), extracts cholesterol from the ER membrane, and transfers it to TGN membranes. The transfer activity consumes PI4P, enriched in TGN membranes produced by PI4-kinases. Depletion of OSBP has a significant impact on protein sorting in the trans-Golgi Network (TGN). This is manifested by missorting different plasma membrane proteins and losing polarity. The authors suggest that OSBP generates lipid gradients in the TGN required for cargo sorting.

The experiments have been carefully performed, and the research question is highly significant and exciting for JCB readers. Nevertheless, there are a few questions that should be clarified. The clarity of the text can also be greatly improved.

A: Thank you for these overall very positive comments.

Major concerns

1. Please clarify: OSBP regulates apical and basolateral sorting in the TGN. Is it surprising that both cargo types are in proximity to OSBP? Are there cargo types that are excluded? Could all cargo proteins move through this domain initially?

A: We performed an unbiased TurboID experiment to explore the proteomic environment of OSBP under different conditions (control and ORPphilin treatments). The ER-Golgi localization of OSBP puts it in a good position to transiently see most plasma membrane proteins (including apical and basolateral) during their trafficking. Therefore, we agree that it is not surprising that most – if not all – plasma membrane proteins meet OSBP. However, this has never been shown before. Moreover, our key result is rather that OSBP is an essential protein for the sorting of these cargoes. Notably, luminal, secreted cargoes were not present in our proteomic dataset, as they are not accessible to the biotin ligase fusion construct used here. Nevertheless, proteomic analyses of cellular secretomes derived from wild type and OSBP-depleted cells help us to complete the analysis by showing that OSBP also regulates the secretion of such soluble proteins.

2. In the MS analysis with OSW treatment, the authors identify more basolateral proteins near OSBP and conclude that apical and basolateral cargo must be segregated in the TGN. However, the segregation hypothesis would require confirmation by immunostaining (apical versus basolateral versus OSBP and/or PI4P).

A: Thank you for this suggestion. We performed the suggested experiment. However, please note that the repertoire of antibodies to detect endogenous proteins is limited as we are using a canine cell line. Typically, we are using CDH1 and GP135 antibodies to detect endogenous basolateral and apical cargoes in MDCK cells. However, the fraction of the given cargo of interest at the Golgi can heavily vary – e.g. GP135 never gives Golgi signal at steady state, whereas CDH1 is usually detected both in the Golgi and the PM.

To overcome this, we used the RUSH system – expressing an apically detected GPI-EGFP model cargo – and combined this with staining of endogenous OSBP and CDH1. First, we transfected the cells with the RUSH cargo-expressing vector, then on the next day we initiated the ER release of the cargo by biotin addition. We accumulated the cargo in the TGN by temperature

block, then we transferred the cells to 37°C in the presence or absence of SWG for 15 min. Following this, cells were fixed and endogenous OSBP and CDH1 was labelled (**Panel A**). In the control cells we detected perinuclear areas positive for either GPI-EGFP or CDH-1. We also identified regions where both cargoes and OSBP were present (**Panel B**). In the SWG-treated cells, a more prominent overlap was detected between endogenous CDH1 and OSBP (see blue and magenta lines), suggesting that SWG directs OSBP to “CDH1-rich” Golgi areas (**Panel C**).

Although this observation is encouraging, we decided not to include these experiments in the main manuscript due to the following reasons:

I. Selection of Golgi areas might be biased. How to address these co-localizations in a more quantitative manner? Image analysis and statistical assessment to measure cell-to-cell and experiment-to-experiment variances is extremely difficult for such experiments.

II. Although we selected cells with low-expression levels for imaging, the fact that we cannot image the endogenous cargo implies that we can still overload the Golgi membranes leading to potential failure of cargo sorting.

III. In order to accumulate cargoes at the TGN, we applied a temperature block and treated the cells with SWG only after the temperature block release. However, at this step, the time window is very narrow to detect Golgi-localized cargoes. SWG induces full OSBP recruitment and PI(4)P accumulation in 30-40 min after addition; therefore, after 15 min, only a milder effect on OSBP recruitment is expected.

3. What is the significance of hits in OSW-treated cells? You generate a huge domain artificially. Since sorting is a highly dynamic process, how can this domain capture cargo molecules precisely? Please explain.

A: We partially agree. As any proximity-based proteomic assay, TurboID gives many hits (including some false positives). Here, we use two drugs that block OSBP in a conformation that stably tethers the ER and TGN. The first reassuring information is the fact that the spectra of interactants and their fold change with SWG and OSW1 are remarkably similar (Fig3E-F). The second reassurance is the decrease in cytosolic hits and the increase in plasma membrane

hits upon treatments, which suggest that the plasma membrane hits that we get without drugs are cargoes crossing the Golgi. But the third and more unexpected finding is that basolateral cargoes are overrepresented, whereas apical ones seems excluded from such artificial domains. This observation must be analyzed in light of what we know about OSBP membrane interaction properties: This includes affinity for PI(4)P, a strong preference for disordered membrane regions and yet a lipid exchange activity that should increase lipid order. When this exchange is blocked, the TGN cannot acquire new membrane features and becomes compatible mostly with basolateral cargoes as judged by the TurboID hits. However, please note that we do not base our reasoning only on the TurboID experiments with drugs. Other approaches (e.g. RUSH assays with apical and basolateral cargoes – Fig2A-D) point out that OSBP targeting affects the trafficking of these cargoes differently.

4. Both apical and basolateral membranes contain cholesterol. Why is the sorting of basolateral proteins much more affected?

A: The effects depend on the strategy (see also previous response). When we silenced OSBP, both apical and basolateral export slowed down, suggesting that OSBP-mediated lipid transfer contributes to the transport of the two types of cargoes. Similarly, when we interrogated the proteins that are close to active OSBP by proximity labeling (TurboID), we found no correlation between the detected cargo proteins and their apical or basolateral character. However, the reviewer is correct that we observed a more striking effect on basolateral cargoes when we blocked OSBP by ORPphilins. Under these conditions, the TGN exit of basolateral cargoes is completely blocked whereas the TGN exit of apical cargoes slowed down as monitored either by RUSH assays or protein surface biotinylation. We interpret the difference between the silencing and pharmacological approaches in the light of the molecular mechanisms by which ORPphilins act. They not only block lipid exchange but also stabilize OSBP in enlarged and stable contact sites where TGN membranes remain PI(4)P-rich and do not acquire cholesterol. Basolateral (and possibly non-polarized) cargoes get concentrated in these regions as revealed by their preferential biotinylation by OSBP-TurboID in the presence of ORPphilins. To conclude, the lipid exchange activity of OSBP is important for both cargoes, but trapping OSBP in an inactive conformation reveals that the TGN regions on which OSBP initially act and which are rich in PI(4)P and poor in cholesterol are more prone to host basolateral than apical cargoes. We have done multiple changes in the text to better emphasize the difference between the silencing and pharmacological approaches and the information we get from their comparison.

5. The authors talk about various secretory routes (line 373). Which routes are they referring to?

A: We apologize for the incorrect phrasing. We were referring to apical and basolateral trafficking routes. This has been corrected in the revised manuscript.

Minor comments and suggestions:

Please change Golgi to TGN, be consistent with the expression - DONE

Please use cargo, not secretory cargo (you look at membrane-associated proteins) - DONE

Please do not use the word secretion for cell surface transport (proteins stay associated with the membrane) - DONE

Line 163to monitor cargo export dynamics... - DONE

Line 165.... inhibit/impair OSBP-mediated lipid exchange.... - DONE

Line 168... Please reformulate the sentence. It is confusing.

At 20 degrees, cargo colocalizes with mcherry-OSBP in control cells... -

The shift to 37 degrees releases cargo from the TGN while OSBP still localizes to the TGN.

In contrast, treatment of cells with OSW significantly impairs TGN export of basolateral cargo at 37 degrees as demonstrated by a persistent co-localization with OSBP over time. - DONE

Line 177...apical and basolateral cargo require OSBP-dependent MCS... - DONE

Line 194 - 195 OSBP-engaged MCS govern apical and basolateral sorting routes.... - DONE

Line 251 domain-selective mass spectrometry. Please describe what it is in a short sentence.
- DONE

Line 273 ...the abundance... - DONE

OSW treatment impaired TGN export of CDH1

Line 340... we directly analyzed.. - DONE

Reviewer #3 (Comments to the Authors (Required)):

This manuscript investigates the relationship between the TGN-localized PI4P/cholesterol exchanger OSBP and the secretory function of the Golgi in polarized epithelial cells. Genetic or pharmacological disruption of OSBP disrupts export of model apical and basolateral cargoes from the TGN (with a more severe effect on basolateral), and also impairs their segregation into distinct secretory carriers. It also impairs successful formation and maintenance of polarized cysts in an MDCK model of epithelia formation. A proximity proteomic screen confirms stalling of both apical and basolateral cargoes, with a more profound effect on the latter. OSBP inhibition also prevents cholesterol enrichment in both apical and basolateral membranes, and increases membrane disorder. Finally, the epithelial phenotype is shown to turnover more PI4P and be associated with higher expression of cholesterol-trafficking proteins ORP2 and OSBP - both of which correlate with increased survival of patients with lung adenocarcinoma, suggesting reduced epithelial-to-mesenchymal transition.

The manuscript is well written and the data are clearly presented. A combination of genetic and pharmacological approaches are used, and an array of candidate and non-biased screens are employed. The data are consistent and compelling. Overall, the requirements for OSBP in polarized secretion and the establishment and maintenance of a polarized epithelium are clear. Mechanistically, the requirements for OSBP cholesterol/PI4P exchange for polarized secretion is hazier. It is clearly required for PM cholesterol maintenance, but a killer experiment showing cholesterol enrichment at the TGN facilitating sorting is not shown (nor can the reviewer envision a feasible experiment to show this). That being said, the observation of OSBP lipid traffic playing a central role in polarized secretion and the epithelial phenotype is an important finding, and one that has significance for a wide swathe of the community. Some suggestions to improve the manuscript are listed below:

A: We are grateful to the Reviewer for providing constructive criticism to improve the quality of the manuscript. We agree with the Reviewer that showing directly how cholesterol enrichment in the Golgi facilitates sorting would be a groundbreaking experiment.

We attempted to detect membrane order of the Golgi membranes directly to investigate how lipid composition of Golgi regions contribute to sorting/trafficking. For this, we used the organelle-selective solvatochromic probes provided by A. Klymchenko (Danylchuk et al. 2021) to image the Golgi under different conditions (Control, ORPphin treatments), similar to that shown in Fig 6 (NR12A-plasma membrane). During these experiments, we encountered specificity issues in MDCK cells, as not only the Golgi, but other intracellular membranes (presumably ER) were also strongly labelled with the Golgi-specific dye. To overcome this, we repeated the experiment using bGalT1-BFP (TGN marker)-expressing cells. Importantly, the BFP emission spectrum does not overlap with the Golgi-specific Nyle Red emission wavelengths; thus, the resulted images could be used to mask the Golgi and to perform ratiometric image analysis of the regions of interest (bGalT1-BFP-positive trans-Golgi regions). We found that the ratio signals were not homogenous throughout the Golgi membranes: 'hot' and 'cold' spots were detected. They might correspond to 'sub-domains' for sorting (See panel below). Encouragingly, we found that pre-treatments with OSW-1 slightly shifted the pixel-ratio values, indicating a potential drop in the cholesterol content of such membranes under OSBP blockage. However, instead of having a more homogenous ratio value distribution (narrower spectra), the previously mentioned heterogeneity in pixel ratio values still remained

(the width of the two spectra are comparable). These are encouraging results. However, the next and very demanding steps are to improve (i) the organelle specificity of the probe; (ii) its sensitivity range to lipid bilayer polarity at the membrane interface; (iii) our ability to image in time and space of the polarity probe in the small TGN region. We believe that these points are beyond the scope of this manuscript.

Conceptual suggestions:

The authors' explanation that OSBP inhibition leads to accumulation of basolateral cargo at the TGN because inhibited, PI4P-bound OSBP accumulates at membrane domains rich in PI4P but poor in cholesterol and disordered seems feasible. My idea is: what if PI4Ks (PI4Kbeta and PI4K2alpha) are also inhibited? Would this not cause OSBP enrichment to stop, and at least equalize the rate of apical and basolateral protein secretion (even though it will likely still block sorting)? Indeed, a requirement for cholesterol in membrane protein sorting at the TGN might explain the key requirement for a TGN/ER cholesterol exchanger when many in the cholesterol field poo-poo the idea in favor of direct lysosome/PM and ER/PM exchange.

A: We agree that complete depletion of the Golgi PI(4)P pool should prevent the accumulation of OSBP at disordered Golgi membranes. Although the proposed experiment is appealing, several problems hinder its feasibility. First, other PI(4)P-effectors have key roles in trafficking. Thus, complete PI(4)P depletion leads to very broad defects on both trafficking and sorting. For example, GOLPH3 binds to PI(4)P to support the formation of Golgi-derived secretory vesicles. We found that PI4KIIIbeta-derived PI(4)P is essential for Golgi-targeting of GOLPH3, as administration of PIK93 triggered the dissociation of GOLPH3-EGFP from Golgi membranes (See figure below).

Nevertheless, we tried to perform the suggested experiment. Although there are good pharmacological tools to target PI4KIIIbeta, we lack specific drugs to inhibit PI4KIIalpha. To bypass this limitation, we used PAO – a pan-PI4-kinase inhibitor, which effectively depletes total cellular PI(4)P production. Unfortunately, this drug is strongly cytotoxic and cells typically die within 15-20 min after addition. Therefore, it was not possible to combine this treatment with the RUSH system and with OSBP inhibiting drugs. Although siRNA-mediated kinase silencing could be a good alternative, the broad effects of complete PI(4)P depletion on cargo trafficking and sorting prompted us to not go further.

The use of PIK93 in the experiments in figure 1H and I is problematic, since this compound is known to inhibit class 1B and class III PI3K, which complicates interpretation of the data. A more selective compound such as PI4KIIIbeta-IN-10 should be employed for these experiments.

A: We thank the reviewer for this suggestion. We repeated some of the experiments using PI4KIIIbeta-IN-10, as suggested. As shown in new figures (Figure S1C-G), this drug depletes Golgi PI(4)P and strongly reduces Golgi-OSBP in MDCK cells. Next, we observed that PI4KIIIbeta-IN-10 treatment reduced the sorting capacity of the MDCK cells as we observed significantly more double positive post-Golgi vesicles upon drug treatment compared to control cells. Last, we developed 3D cysts in the presence of 25 nM PI4KIIIbeta-IN-10 and observed a significant increase in the number of cysts with morphology defects. These results, which strengthen our hypothesis, are now incorporated in the revised form of the manuscript.

Comments on technical aspects:

The methods section makes clear that 500 nM PIK93 was used, but this should be detailed in the figure legends for convenience (e.g. figure 1)

A: Done. The concentration of PIK93 is now incorporated in the corresponding section of Figure Legend 1.

The number of independent experiments is not clear in figure 1J-K, 2A-D and should be stated.

A: In the revision, we now incorporated all three experiments, showing one representative experiment in the main figure and all the 3 repeats side-by-side in the supporting information. Sometimes the amplitude of the OSBP recruitment, the time of the ER to Golgi transfer or the slope of Golgi exit rate varied among the repeats, making it difficult to bring each experiment together into one single plot. However, the repeated experiments gave consistently the same results (delayed apical cargo export vs complete block of basolateral cargo export).

Figures 3C, 4 E, F, H: Welch's t-test is used, but there are multiple comparisons. The data should be compared by one-way ANOVA and a suitable post-hoc comparison test

A: Thank you for this suggestion. In the revision, the statistical tests for the above-mentioned experiments were performed using a one-way ANOVA and a Dunnett's post-hoc test. The new P-values have been changed on the corresponding Figures, and the statistical method used has been added to the figure legends. Similar to the t-tests used previously, the one-way ANOVA tests found significant differences across the mean values examined.

In addition to this, we repeated the statistical analysis corresponding to the RUSH tests, live imaging tests and to the experiments shown on Figure 1 G and H. Since upon those experiments we determine how the response is affected by two factors (time and treatment, vesicle category and treatment, plasma membrane domain and treatment) we changed the multiple t-tests to two-way ANOVA with Sidak's multiple comparisons, respectively.

Figure 5D-F: the manuscript states that the histograms show means of 4-fields with 8-10 cells. However, the number of independent experiments analyzed is not clear. It reads like all four fields could have come from the same experiment

A: The experiments shown in Figure 5 illustrate one representative measurement (from the image per se, to the quantification of the wavelength shift). The experiments were repeated at least three times, and gave similar outcomes. We included all the performed experiments in the revised version of the manuscript as supplementary material.

Figure S1A: details of mean, error and n are missing from the legend.

A: We agree. Statistical details have been added to Figure S1A and to the corresponding figure legend.

October 20, 2023

RE: JCB Manuscript #202307051R

Dr. Bruno Antony
Université Côte d'Azur et CNRS
Institut de Pharmacologie Moléculaire et Cellulaire
660 route des Lucioles
Valbonne 06560
France

Dear Dr. Antony:

Thank you for submitting your revised manuscript entitled "Lipid exchange at ER-trans Golgi contact sites governs polarized cargo sorting". We would be happy to publish your paper in JCB pending final revisions necessary to meet our formatting guidelines (see details below).

A. MANUSCRIPT ORGANIZATION AND FORMATTING:

- 1) Text limits: Character count for Articles is < 40,000, not including spaces. Count includes abstract, introduction, results, discussion, and acknowledgments. Count does not include title page, figure legends, materials and methods, references, tables, or supplemental legends.
- 2) Figures limits: Articles may have up to 10 main text figures.
- 3) * Figure formatting: Scale bars must be present on all microscopy images, including inset magnifications (you may alternatively indicate the diameter of the inset). Molecular weight or nucleic acid size markers must be included on all gel electrophoresis. *
- 4) Statistical analysis: Error bars on graphic representations of numerical data must be clearly described in the figure legend. The number of independent data points (n) represented in a graph must be indicated in the legend. Statistical methods should be explained in full in the materials and methods. For figures presenting pooled data the statistical measure should be defined in the figure legends. Please also be sure to indicate the statistical tests used in each of your experiments (either in the figure legend itself or in a separate methods section) as well as the parameters of the test (for example, if you ran a t-test, please indicate if it was one- or two-sided, etc.). Also, if you used parametric tests, please indicate if the data distribution was tested for normality (and if so, how). If not, you must state something to the effect that "Data distribution was assumed to be normal but this was not formally tested."
- 5) Abstract and title: The abstract should be no longer than 160 words and should communicate the significance of the paper for a general audience. The title should be less than 100 characters including spaces. Make the title concise but accessible to a general readership.
- 6) Materials and methods: Should be comprehensive and not simply reference a previous publication for details on how an experiment was performed. Please provide full descriptions in the text for readers who may not have access to referenced manuscripts.
- 7) Please be sure to provide the sequences for all of your primers/oligos and RNAi constructs in the materials and methods. You must also indicate in the methods the source, species, and catalog numbers (where appropriate) for all of your antibodies. Please also indicate the acquisition and quantification methods for immunoblotting/western blots.
- 8) Microscope image acquisition: The following information must be provided about the acquisition and processing of images:
 - a. Make and model of microscope
 - b. Type, magnification, and numerical aperture of the objective lenses
 - c. Temperature
 - d. Imaging medium
 - e. Fluorochromes

- f. Camera make and model
 - g. Acquisition software
 - h. Any software used for image processing subsequent to data acquisition. Please include details and types of operations involved (e.g., type of deconvolution, 3D reconstitutions, surface or volume rendering, gamma adjustments, etc.).
- 9) References: There is no limit to the number of references cited in a manuscript. References should be cited parenthetically in the text by author and year of publication. Abbreviate the names of journals according to PubMed.
- 10) Supplemental materials: There are strict limits on the allowable amount of supplemental data. Articles may have up to 5 supplemental figures. Please also note that tables, like figures, should be provided as individual, editable files. A summary of all supplemental material should appear at the end of the Materials and methods section. Please ensure to include a legend for all videos.
- 11) eTOC summary: A ~40-50-word summary that describes the context and significance of the findings for a general readership should be included on the title page. The statement should be written in the present tense and refer to the work in the third person.
- 12) Conflict of interest statement: JCB requires inclusion of a statement in the acknowledgements regarding competing financial interests. If no competing financial interests exist, please include the following statement: "The authors declare no competing financial interests." If competing interests are declared, please follow your statement of these competing interests with the following statement: "The authors declare no further competing financial interests."
- 13) ORCID IDs: ORCID IDs are unique identifiers allowing researchers to create a record of their various scholarly contributions in a single place. Please note that ORCID IDs are now *required* for all authors. At resubmission of your final files, please be sure to provide your ORCID ID and those of all co-authors.
- 14) A separate author contribution section following the Acknowledgments. All authors should be mentioned and designated by their full names. We encourage use of the CRediT nomenclature.

Please note that JCB now requires authors to submit Source Data used to generate figures containing gels and Western blots with all revised manuscripts. This Source Data consists of fully uncropped and unprocessed images for each gel/blot displayed in the main and supplemental figures. Since your paper includes cropped gel and/or blot images, please be sure to provide one Source Data file for each figure that contains gels and/or blots along with your revised manuscript files. File names for Source Data figures should be alphanumeric without any spaces or special characters (i.e., SourceDataF#, where F# refers to the associated main figure number or SourceDataFS# for those associated with Supplementary figures). The lanes of the gels/blots should be labeled as they are in the associated figure, the place where cropping was applied should be marked (with a box), and molecular weight/size standards should be labeled wherever possible.

Journal of Cell Biology now requires a data availability statement for all research article submissions. These statements will be published in the article directly above the Acknowledgments. The statement should address all data underlying the research presented in the manuscript. Please visit the JCB instructions for authors for guidelines and examples of statements at (<https://rupress.org/jcb/pages/editorial-policies#data-availability-statement>).

B. FINAL FILES:

****It is JCB policy that if requested, original data images must be made available to the editors. Failure to provide original images upon request will result in unavoidable delays in publication. Please ensure that you have access to all original data images prior to final submission.****

****The license to publish form must be signed before your manuscript can be sent to production. A link to the electronic license to publish form will be sent to the corresponding author only. Please take a moment to check your funder requirements before choosing the appropriate license.****

Thank you for this interesting contribution, we look forward to publishing your paper in Journal of Cell Biology.

Sincerely,

William Prinz, PhD
Monitoring Editor

Andrea L. Marat, PhD
Senior Scientific Editor

Journal of Cell Biology